# IL-17B/RB Activation in Pancreatic Stellate Cells Promotes Pancreatic Cancer Metabolism and Growth

**DOI:** 10.3390/cancers13215338

**Published:** 2021-10-24

**Authors:** Jiahui Li, Xiaolin Wu, Lars Schiffmann, Thomas MacVicar, Chenghui Zhou, Zhefang Wang, Dai Li, Oscar Velazquez Camacho, Reiner Heuchel, Margarete Odenthal, Axel Hillmer, Alexander Quaas, Yue Zhao, Christiane J. Bruns, Felix C. Popp

**Affiliations:** 1Department of General, Visceral, Tumor and Transplantation Surgery, University Hospital Cologne, Kerpener Straße 62, 50937 Cologne, Germany; jiahuiliss@163.com (J.L.); xiaolin.wu@uk-koeln.de (X.W.); lars.schiffmann@uk-koeln.de (L.S.); chenghui.zhou@uk-koeln.de (C.Z.); buxiaodao@163.com (Z.W.); lidaichhosp@163.com (D.L.); yue.zhao@uk-koeln.de (Y.Z.); 2Max-Planck-Institute for Biology of Ageing, 50931 Cologne, Germany; tmacvicar@age.mpg.de; 3Department of Plastic and Reconstructive Surgery, Second Affiliated Hospital, School of Medicine, Zhejiang University, Hangzhou 310052, China; 4Department of Anesthesiology, Changhai Hospital, Naval Medical University, Shanghai 200433, China; 5Institute of Pathology, University of Cologne, 50937 Cologne, Germany; oscar.velazquez-camacho@uk-koeln.de (O.V.C.); margarete.odenthal@uk-koeln.de (M.O.); ahillmer@uni-koeln.de (A.H.); alexander.quaas@uk-koeln.de (A.Q.); 6Pancreas Cancer Research Lab, Department of Clinical Intervention and Technology (CLINTEC), Karolinska Institute, 171 77 Stockholm, Sweden; rainer.heuchel@ki.se; 7Center for Molecular Medicine (CMMC), University of Cologne, 50931 Cologne, Germany

**Keywords:** pancreatic cancer, tumor microenvironment, IL17B/RB, metabolism

## Abstract

**Simple Summary:**

Pancreatic cancer has the lowest survival rate of all malignancies. Understanding the interplay between tumor and stroma could lead to the development of new therapies. The metabolic role of interleukin 17B/interleukin 17B receptor (IL-17B/RB) has not been adequately studied in pancreatic cancer and is poorly understood. Here, we investigate the IL-17B/RB-mediated interactions between the tumor and the stroma. We analyze murine as well as human stromal and tumor cells, animal experiments with immunocompromised mice, and human cell lines with overexpression and knockdown of IL-17RB. We report that aberrant expression of IL-17B/RB in stromal pancreatic stellate cells (PSCs) accelerates tumor cell growth. IL-17B/RB-signaling supplies energy by increased oxidative phosphorylation (OXPHOS). Blocking IL-17B/RB to inhibit the tumor to stroma crosstalk could be a potential targeted therapy for pancreatic cancer.

**Abstract:**

In pancreatic ductal adenocarcinoma (PDAC), the tumor stroma constitutes most of the cell mass and contributes to therapy resistance and progression. Here we show a hitherto unknown metabolic cooperation between pancreatic stellate cells (PSCs) and tumor cells through Interleukin 17B/Interleukin 17B receptor (IL-17B/IL-17RB) signaling. Tumor-derived IL-17B carrying extracellular vesicles (EVs) activated stromal PSCs and induced the expression of IL-17RB. PSCs increased oxidative phosphorylation while reducing mitochondrial turnover. PSCs activated tumor cells in a feedback loop. Tumor cells subsequently increased oxidative phosphorylation and decreased glycolysis partially via IL-6. In vivo, IL-17RB overexpression in PSCs accelerated tumor growth in a co-injection xenograft mouse model. Our results demonstrate a tumor-to-stroma feedback loop increasing tumor metabolism to accelerate tumor growth under optimal nutritional conditions.

## 1. Introduction

Pancreatic cancer has the lowest survival rate of all cancers and is predicted to become the 2nd leading cause of cancer death by 2030 [1]. Its characteristic desmoplastic stroma contributes to pancreatic cancer’s dismal prognosis and therapeutic challenges [2,3]. Pancreatic stellate cells (PSCs) constitute a major stromal component and form a niche for cancer stem cells [4]. Upon activation, under hypoxic conditions, PSCs differentiate into cancer-associated fibroblasts (CAFs) [5]. Activated PSCs and CAFs express alpha-smooth muscle actin (α-SMA) and secrete extracellular molecules creating a physical barrier facilitating chemoresistance [6]. PSCs also support tumor metabolism by providing nutrients, for example, alanine secretion in an autophagic process [7]. Here we ask if PSCs interact with pancreatic cancer cells through Interleukin 17B/Interleukin 17B receptor (IL-17B/IL-17RB) cell-cell signaling. Wu et al. described an autocrine feedback loop of IL-17B/IL-17RB signaling that activates multiple chemokines to enhance pancreatic cancer malignancy [8]. The inhibition of IL-17B using shRNA or treatment with IL-17B blocking antibody reduced pancreatic tumor growth and prevented metastasis formation in a mouse xenograft model. IL-17B/IL-17RB signaling also increases the expression of the chemokines Chemokine (C-C motif) ligand 20 (CCL20), C-X-C Motif Chemokine Ligand 1 (CXCL1), Interleukin 8 (IL-8), and Trefoil factor 1 (TFF1) via the extracellular signal-regulated protein kinase (ERK1/2) pathway. These chemokines recruit macrophages and endothelial cells into the tumor and are crucial for the persistence of pulmonary metastasis [8]. IL-17B contributes to metastasis in lung cancer cells in a xenograft mouse model [9] and enhances the stemness of gastric cancer cells [10].

We hypothesize that a tumor-promoting interaction of PSCs and tumor cells via IL-17B/IL-17RB may affect energy metabolism. Increased energy production would lead to tumor growth and would have to be mediated by mitochondria. Mitochondria are organelles that generate energy in the form of adenosine triphosphate (ATP) through oxidative phosphorylation (OXPHOS). Mitochondria undergo fission and fusion processes in response to changes in energy demand and cellular stress. These processes may also play a role in tumorigenesis. The three main steps of mitochondrial fusion are tethering of two mitochondria, docking of the membranes, and fusion of the outer mitochondrial membranes (OMM). Fusion of OMM is induced by two GTPases, mitofusin1 (MFN1) and mitofusin2 (MFN2) [11]. The GTPase dynamin-related protein 1 (Drp1) plays an essential role in mitochondrial fission [12,13]. Among the organelles that can be degraded by autophagy are mitochondria. The selective degradation of mitochondria by autophagy is referred to as mitophagy [14]. It has been demonstrated that extracellular lactate can promote tumor mitochondrial energy production. One of the theories describing the metabolic coupling between the tumor cell and cancer-associated fibroblasts (CAFs) is the reverse Warburg effect. Here, tumor-induced glycolysis in the CAF supplies adjacent cancer cells with lactate, inducing the tricarboxylic acid (TCA) cycle for ATP generation. This increases tumor proliferation and decreases cell death.

It remains an open question if and how IL-17B mediates tumor-to-stroma signaling in pancreatic cancer. In particular, the investigation of the interaction between tumor cells and PSCs seems to be promising. PSCs are the principal source of stromal collagen [10] and can promote tumor growth [7]. However, the stroma does not generally support tumor development—the complete depletion of the stroma leads to the progression of pancreatic cancer [15,16]. If IL-17B mediates the crosstalk between PSCs and tumor cells, the question is whether IL-17B promotes tumor growth. In this work, we investigated the interaction of PSCs with pancreatic cancer cells with particular emphasis on tumor metabolism.

## 2. Materials and Methods

### 2.1. Human ‘Specimens’ Collection

Tumors, corresponding non-tumor tissues, and serum were collected from patients diagnosed with pancreatic ductal adenocarcinoma at the University Hospital of Cologne, Department of General, visceral, tumor, and transplant surgery. Patients’ consent was obtained for the current study, which was approved by the Ethics Committee of the University of Cologne. Serum samples were also collected from healthy volunteers with informed consent.

### 2.2. Mitostress and Glycolytic Rate Assay

Oxygen consumption rate (OCR) and glycolytic Proton Efflux Rate (glycoPER) were measured with the XF96 extracellular flux analyzer (Agilent, Santa Clara, CA, USA). In brief, human PSCs or PDAC cells were seeded on XF96 plates at 0.25 × 10^6^ cells/well in 180 μL XF base medium (#102353-100, Agilent, CA, USA), supplemented with 10 mM D-glucose (G6152, Sigma-Aldrich, St. Louis, MO, USA), 1 mM sodium pyruvate (#11360088, Thermo Fisher Scientific, Waltham, MA, USA), 4 mM L-glutamine (#25030081, Thermo Fisher Scientific, Waltham, MA, USA) and adjusted to pH 7.4. Live OCR was measured under basal conditions and mitochondrial inhibitors (1 μM oligomycin, O4876, Sigma-Aldrich, St. Louis, MO, USA), 1.5 μM of Carbonyl cyanide 4-(trifluoromethoxy) phenylhydrazone (FCCP, C2920, Sigma-Aldrich, St. Louis, MO, USA), and 1 μM rotenone (R8875, Sigma-Aldrich, St. Louis, MO, USA) combined with 2 μM antimycin A (A8674, Sigma-Aldrich, St. Louis, MO, USA). Live glycoPER was measured under basal conditions and mitochondrial and glycolytic inhibitors (1 μM rotenone, R8875, Sigma-Aldrich, St. Louis, MO, USA combined with 2 μM antimycin A, A8674, Sigma-Aldrich, St. Louis, MO, USA, and 50 μM 2-DG, D6134, Sigma-Aldrich, St. Louis, MO, USA).

### 2.3. Mitochondria Isolation

After washing twice with PBS, tumor cells or PSCs were resuspended in homogenization buffer containing 70 mM sucrose, 220 mM mannitol, 1 mM EGTA (Roche, Mannheim, Germany), and 5 mM HEPES/KOH (pH 7.4) together with complete protease inhibitor (Roche, Mannheim, Germany). A rotating Teflon potter was applied at 1000 rpm to homogenize the cell suspension. After removing cell debris and nuclei with centrifugation twice at 600× *g* for 5 min at 4 °C, the mitochondrial fraction was obtained from the supernatant by centrifugation at 8000× *g* for 10 min at 4 °C.

### 2.4. Electronic Microscope

Transmission electron microscopy was applied to assess the morphology of the PSC mitochondria. Briefly, cells were fixed with a primary fixation buffer containing 2% paraformaldehyde and 2.5% glutaraldehyde in a 0.1 M phosphate buffer. Then, cells were fixed in the secondary fixation buffer with 1% osmium tetroxide and 1.5% potassium ferrocyanide in a 0.1 M phosphate buffer. After embedding in low melting agarose, sections of the cell blocks were dehydrated and embedded with EPON 812 resin. Finally, ultrathin sections were attached to uncoated nickel grids and contrasted with aqueous uranyl acetate and lead citrate. Images were taken with a Zeiss EM912 Omega equipped with a TRS 2K bwCCD-camera.

### 2.5. ELISA

Human IL-6 (RAB0306, Thermo Fisher, Waltham, MA, USA), IL-17B (DY1248, R&D Systems, Minneapolis, MN, USA) ELISA kits were used to measure the IL-6 and IL-17B concentration. The analysis was performed in cell culture supernatants as well as in human pancreatic tissue and serum exosomes (IL-17B) as per the manufacturer’s instructions. All samples were measured in duplicate and compared with a standard curve to determine the concentration.

### 2.6. Exosome Preparation

Exosomes were prepared from human serum and cell culture supernatants. For cell culture supernatants, a serum-free culture medium was applied for 48 h before the exosome isolation. In brief, whole cells, dead cells, and cell debris were removed by serial centrifugation at 300× *g*, 10 min; 2000× *g*, 10 min; 10.000× *g*, 30 min at 4 °C. Exosomes were resuspended after ultracentrifugation at 100.000× *g*, 70 min at 4 °C (OptimaTM L-90K, Beckman Coulter, Brea, CA, USA). Human serum exosomes were isolated with the ExoQuick kit (EQ806A-1, System Biosciences, Palo Alto, CA, USA) according to the manufacturer’s instructions.

### 2.7. In Vivo Experiments

The Institute of Animal Care and Welfare of the University of Cologne and regional authorities approved all animal experiments (81-02.04.2018. A139, LANUV NRW). For the Xenograft model, 2 × 10^5^ MIA PaCa-2 cells were injected alone or together with 1 × 10^6^ IL-17RB overexpressing PSCs (OE) respective control PSCs (VEC OE) in the flanks of NOD-scid IL-2Rgamma^null^ mice (NSG, CECAD, University of Cologne, Cologne, Germany). Tumor diameters were measured every second day, and the tumor volume was calculated with the formula (width^2^ × length)/2. Tumor volumes were measured with a digital caliper. All mice were sacrificed when one tumor’s maximum diameter reached 15 mm or on day 46 under anesthesia. The final tumor size was calculated as (width × length × height) × 0.5236 with a digital caliper.

### 2.8. Statistical Analysis

Statistical analysis was performed with R, R-Studio (RStudio PBC, Boston, MA, USA), and GraphPad prism (version 7; GraphPad Software Inc., San Diego, CA, USA). The Mann–Whitney U test and the *t*-test were used to compare two groups as indicated in the figure legends. Analyses were considered significant when the *p*-value was <0.05 (* *p* < 0.05, ** *p* < 0.01).

### 2.9. Publicly Available Data Analysis

Human PDAC single-cell RNA-seq data were obtained from the Genome Sequence Archive (Project number PRJCA001063) [17]. We reanalyzed IL-17B expression (Figure 1D) according to the method described by Dominguez et al. [18]. Expression levels were normalized to the interval from 0 to 1.

IL-17B expression of 24 PDAC patients was analyzed in different cell types using BBrowser 2 (https://bioturing.com/, accessed on 23 October 2021). The online tool GEPIA 2 (http://gepia2.cancer-pku.cn, accessed on 23 October 2021) was used to access the Cancer Genome Atlas (TCGA) and the Genotype-Tissue Expression (GTEx) project to analyze differential expression of IL-17B in tumor and normal tissue.

### 2.10. Pancreatic Stellate Cell Isolation and Immortalization

Primary PSCs were isolated from mouse and human pancreatic tissue samples as previously described with minor modifications [19]. Briefly, after harvesting the pancreas from 6-week-old C57BL/6J mice, the tissue was minced and digested with 0.05% Collagenase P (Roche, Germany) and 0.1% DNase (Roche, Germany) in Gey’s balanced salt solution (GBSS; Sigma Aldrich, CA, USA) at 37 °C for 7 min two times. After filtering through a 100 µm nylon mesh, pellets were washed with GBSS containing 0.3% BSA and resuspended with 28.7% Nycodenz solution, and underwent density gradient centrifugation at 1400× *g* for 20 min at 4 °C. The interface was collected and washed with GBSS and seeded in tissue culture dishes at 37 °C in an incubator. Human PSCs were isolated from PDAC surgical specimens from normal pancreatic tissue adjacent to the tumor using the method described above. For stimulations of PSCs with IL-17B, higher cell numbers were needed. SV40 large T antigen encoded by a plasmid were transfected into primary PSCs using a lentivirus to immortalize cells [20]. Immortalization allowed the culture of large cell numbers. PSCs were used within 10 passages.

### 2.11. Immunohistochemical Staining

Pancreatic cancer specimens from patients undergoing surgery were sliced into 4 mm sections. Primary antibodies (anti-αSMA, 1:200, Abcam, Cambridge, UK; anti-CXCR4, 1:200, Abcam, Cambridge, UK; Anti-IL-17RB, 1:200, ProteoTech, Attendorn, Germany) were incubated overnight at 4 °C and stained with EnVision System-HRP Labeled Polymer Anti-Rabbit (1:1000, Dako, Jena, Germany) according to the manufacturer’s instructions. Double staining was performed using the Double Stain IHC Kit (Abcam, Cambridge, UK) according to the manufacturer’s manual.

### 2.12. Immunofluorescence

PSCs or tumor cells were seeded in eight-well slide dishes (Ibidi, Gräfelfing, Germany) at 5 × 10^4^/well and incubated for 24 h. Then cells were fixed with 4% formaldehyde and blocked with normal horse serum for 30 min. Slides were incubated with primary antibody (anti-αSMA, 1:200, Abcam, Cambridge, UK; anti-IL-17RB, 1:200, ProteoTech, Attendorn, Germany; anti-vimentin, 1:200, Cell Signaling Technology, Danvers, MA, USA; anti-ATP5β, 1:500, Abcam, Cambridge, UK; anti-Tom20, 1:400, Abcam, Cambridge, UK) overnight at 4 °C. Subsequently, cells were incubated with Alexa 546-conjugated anti-mouse IgG (Thermo Fisher Scientific, Waltham, MA, USA) or Alexa 546-conjugated anti-rabbit IgG (Thermo Fisher Scientific, Waltham, MA, USA) for 1 h at room temperature. Finally, nuclear DNA was counterstained with DAPI (4′,6-diamidino-2-phenylindole, 1 mg/mL) for 15 min. After washing with PBS, images were acquired using a fluorescence microscope (to generate Figure 1C, InCellis Cell Imager, Bertin Instruments, Montigny-le-Bretonneux, France) or a confocal microscope (to generate Figure 6A, Zeiss Meta 710, Zeiss, Jena, Germany).

### 2.13. Sphere Formation Assay

After receiving exosomes at a concentration of 100 ng/mL from IL-17RB overexpressing or empty vector control cells, L3.6pl and HPAF-II cells were cultured at a density of 3000 cells/well in ultra-low attachment six-well plates (Corning, Corning, NY, USA) with DMEM/F-12 medium (Thermo Fisher Scientific, Waltham, MA, USA) supplemented with 20 ng/mL human EGF (PeproTech, Hamburg, Germany), 10 ng/mL human FGF (PeproTech, Hamburg, Germany), 5 µg/mL Insulin (PeproTech, Hamburg, Germany), 1% B27 (Thermo Fisher Scientific, Waltham, MA, USA) and 0.4% BSA at 37 °C in a humidified incubator with 5% CO_2_. After 8 days, a phase-contrast microscope was used to count spheres with a diameter over 50 μm in each well.

### 2.14. Western Blotting

Protein was extracted from cultivated cells with RIPA buffer, separated by SDS-PAGE, and then transferred to PVDF membranes. Then antibodies were applied per the manufacturer’s instructions (Mitophagy Antibody Sampler Kit #43110, Mitochondrial Dynamics Antibody Sampler Kit #48799, and Glycolysis Antibody Sampler Kit #8337 Cell Signaling Technology, Danvers, MA, USA) or antibodies against IL-17RB (ProteoTech, Attendorn, Germany, 1:1000), Actin (Abcam, Cambridge, UK, 1:5000) were used to probe proteins. In addition, HRP-conjugated sheep anti-mouse/Rabbit IgG antibodies (1:5000, Cell Signaling Technology, Danvers, MA, USA) were applied to visualize protein expressions with the Pierce ECL Western blotting substrate (Thermo Fisher Scientific, Waltham, MA, USA).

### 2.15. FACS

For the preparation of PDAC cells, a 70 μm cell strainer was applied to filter cell suspension. To evaluate CXCR4 expression, PDAC cells were stained with a Brilliant Violet 510™ anti-human CXCR4 antibody (Biolegend, San Diego, CA, USA) at a concentration of 5 μL per million for 45 min at 4 °C. The labeled cells were analyzed with an LSRFortessa flow cytometer (Beckman Coulter, Brea, CA, USA). Data were analyzed with FlowJo software (BD Life Sciences, Franklin Lakes, NJ, USA). 

### 2.16. Cell Culture

The human pancreatic cancer cell lines MiaPaCa-2, Panc-1, and HPAF-II were purchased from the American Type Culture Collection (ATCC). The immortalized pancreatic stellate cell line RLT-PSC was kindly provided by Reiner Heuchel and Martin Löhr (Karolinska-Institute, Stockholm, Sweden) and has been previously described [20]. The human pancreatic cancer cell line L3.6pl was kindly provided by Dr. G.E. Gallick (University of Texas MD Anderson Cancer Center, Houston, TX) [21]. Two human PSC primary preparations (hPSC1 and hPSC2) were derived from pancreatic cancer patients from the university hospital cologne. The patients gave their formal consent.

For co-culture experiments (Figure 1B), 1 × 10^5^ primary PSC preparations (mPSC = murine PSCs and hPSC = human PSCs) were seeded into the lower chamber of a transwell cell culture system. 1 × 10^5^ pancreatic cancer cells (L3.6pl for human culture and Panc02 for murine) were seeded into the upper chamber. Cells were cultured in DMEM/f12 with 10% FBS for 72 h before harvesting the PSCs for Western blot analysis. We isolated primary murine PSCs (mPSC) from the pancreas of healthy C57BL/6 mice by gradient centrifugation. Primary human PSCs were obtained from pancreatic surgery specimens. We used cells of passage 3–8 for these experiments.

Recombinant human IL-17B treatment: Cells were plated at 2 × 10^5^ cells/well in 6-well plates. The next day, cells were starved for 6 h before applying recombinant human or mouse IL-17B protein (Proteintech, Rosemont, IL, USA) at 200 or 400 ng/mL as indicated for 48 h. 

We prepared the conditioned medium (CM) from murine PDAC cell line Panc02 and human PDAC cell line L3.6pl. For this purpose, we cultured the tumor cells for 48 h to a confluence of 70–90%. Then, we centrifuged the CM twice at 4 °C for 10 min and passed the CM through a 20-μm cell strainer to remove the live cells.

### 2.17. MTT

The MTT assay to assess cell proliferation starts at 20–30% and ends around 80–90% cell confluency. This setup ensures the analysis of cell proliferation and not viability. The MTT Assay on PDAC cells treated with PSC supernatant was performed in a 96-well plate. After 48 h of treatment, 5 mM MTT reagent (Biomol, Hamburg, Germany) was applied to each well and incubated at 37 °C for 3.5 h. Absorbance was quantified at 570 nm with an ELISA reader. 

### 2.18. Lentivirus-Mediated IL-17RB Knockdown and Overexpression

Short hairpin RNA (shRNA) targeting IL-17RB or a control sequence was inserted in the backbone plasmid (Addgene, Watertown, MA, USA) with a tet-on inducible expression system. The shRNA sequences are as follows: shNTC #RHS4743; shIL-17RB 5′-CCCTTCCATGTCTGTGAATTT-’3’ (Thermo Scientific, Waltham, MA, USA). The constructed plasmids were sequenced for qualification. Overexpression of IL-17RB plasmid was designed and purchased from a company (Cyagen, Santa Clara, CA, USA). pLenti CMV/TO SV40 small + Large T (w612-1) was a gift from Eric Campeau (Addgene plasmid #22298; http://n2t.net/addgene:22298, accessed on 23 October 2021; RRID: Addgene_22298). A total of 2.3 mg of each respective plasmid was co-transfected with the envelope PMD2.G (2 mg) and packaging PsPAX2 (4.7 mg) plasmids (Addgene, Watertown, MA, USA) into 293FT cells using Lipofectamine 2000 (Thermo Fisher Scientific, Waltham, MA, USA). RLT-PSC, hPSC1, 2 were incubated with virus supernatant and polybrene for 24 h. Knockdown cells were further selected with 1.5 mg/mL puromycin.

In this study, different backbones were used for the overexpression (OE) and knockdown (KD) of IL-17RB. Thus, we expect differences in protein expression, for example, between the empty vectors (overexpression—VEC OE and knockdown—VEC KD). In the experiments, only the VEC OE group can be compared with the OE group and the VEC KD group with the KD group. Comparison of the VEC OE and VEC KD groups is not reasonable because of the different backbones.

## 3. Results

### 3.1. Pancreatic Cancer Induces IL-17RB Expression in Pancreatic Stellate Cells (PSCs)

To investigate the spatial distribution of IL-17RB expression in pancreatic ductal adenocarcinoma, we determined the expression of IL-17RB and α-SMA in human PDAC samples using immunohistochemistry. Morphologically clearly identifiable tumor cells (white arrows), as well as α-SMA positive stromal cells (black arrows), expressed IL-17RB (Figure 1A). We performed co-culture of PSCs with pancreatic cancer cells and evaluated the IL-17RB expression on PSCs using immunoblots. We used one murine pancreatic stellate cell line (mPSC) and two human primary PSC cultures obtained from pancreatic cancer surgery samples (hPSC1 and hPSC2). Indirectly co-cultured murine PSCs with murine tumor cells expressed more IL-17RB than naïve PSCs. Human PSCs likewise expressed more IL-17RB when co-cultured with human tumor cells compared to naïve PSCs (Figure 1B, Appendix A). To further confirm IL-17RB induction in PSCs, we investigated IL-17RB expression in human and murine PSCs with immunofluorescence microscopy after being treated with pancreatic cancer-derived conditioned medium (CM). We found a higher expression of IL-17RB in CM treated PSCs compared to naive PSCs (Figure 1C). Thus, soluble factors from specific murine and human tumor cell lines stimulate PSCs to upregulate IL-17RB expression.

We hypothesized that pancreatic cancer cells stimulate the upregulation of IL-17RB expression in PSCs through IL-17B. Using publicly available single-cell RNA sequencing data, we analyzed IL-17B expression. Cancer-associated fibroblasts (CAFs) in the tumor microenvironment expressed considerably more IL-17B than fibroblasts (Figure 1D). These stromal cell populations contain the PSCs. Analysis of publicly available TCGA and GTEx data showed higher IL-17B mRNA expression in the tumor than in normal tissue (Figure 1E). We quantified IL-17B in human PDAC compared with the adjacent non-tumor pancreatic tissue using ELISA. The IL-17B concentration in the tumor tissue was significantly higher at the protein level (Figure 1F), showing that IL-17B is present in pancreatic cancer.

We tested whether IL-17B induces the expression of IL-17RB by stimulating human immortalized PSCs with rhIL-17B. In this experiment, rhIL-17B did not increase IL-17RB noticeably. Also, the expression of the PSC activation marker α-SMA did only change upon stimulation with high IL-17B concentrations (400 ng/mL, Figure 1G). However, we identified tumor cell derived extracellular vesicles (EVs) as inducers of increased IL-17RB expression in PSCs (Figure 1H) that carry IL-17B (Figure 1I). Exosomes from pancreatic cancer patients contained more IL-17B compared with healthy controls (Figure 1J), suggesting that IL-17B-carrying EVs upregulate IL-17RB expression in PSCs.

**Figure 1 cancers-13-05338-f001:**
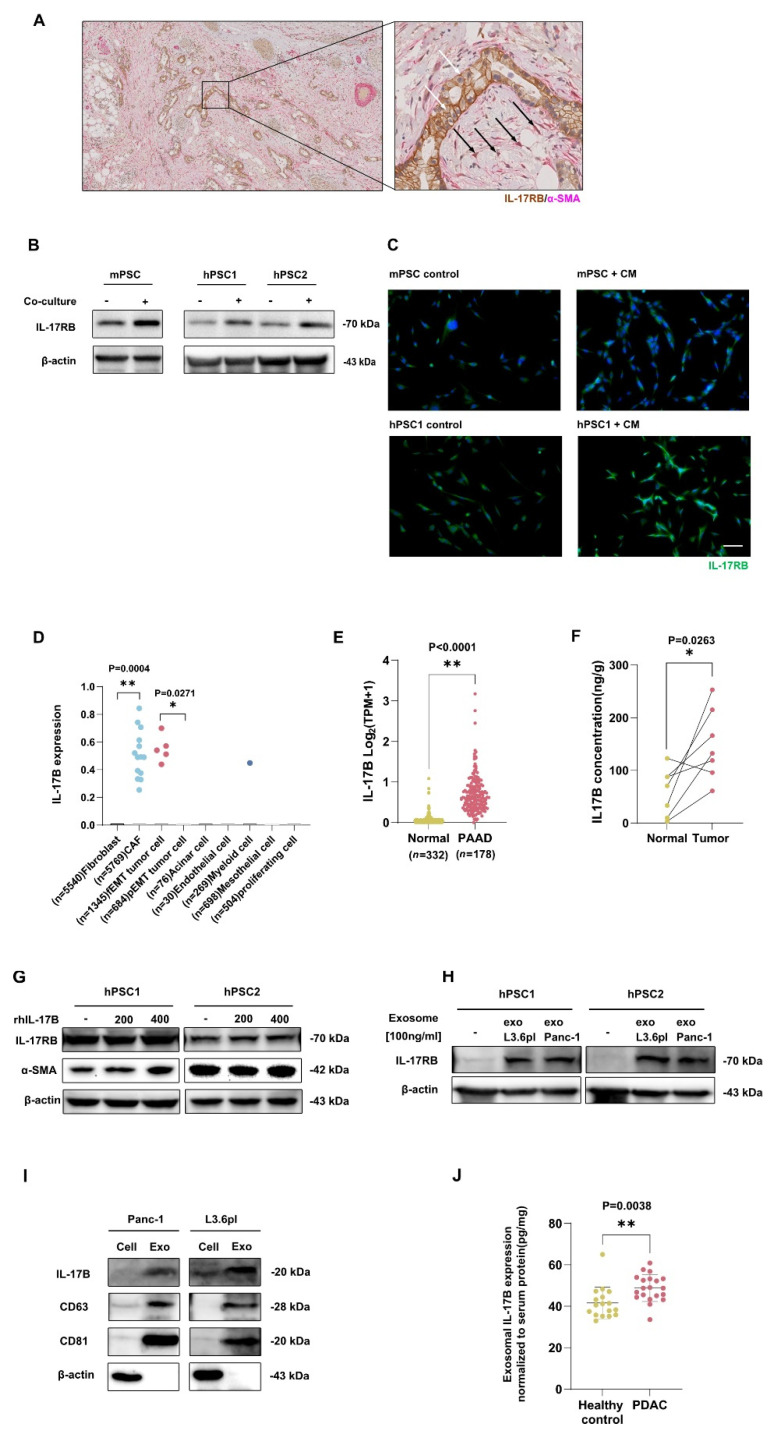
Expression of IL-17RB in PDAC tumors and stroma from humans and mice. (**A**) Double-immunohistolochemistry staining of IL-17RB (brown) and α-SMA. White arrows point to IL-17RB positive tumor cells. Black arrows depict double-positive stromal cells. Scale bars = 100 μm; (**B**) Murine PSCs express more IL-17RB after co-culture with Panc02 pancreatic cancer cells than naive PSCs (left). Two human primary PSCs preparations (hPSC1 and hPSC2) express more IL-17RB after co-culture with L3.6pl pancreatic cancer cells than naive PSCs (right). Western blots of mouse PSCs (RLT-PSCs) and two primary human PSC preparations from surgical specimens. Data show typical images from three independent experiments; (**C**) PSCs increase IL-17RB expression when treated with conditioned medium (CM) from tumor cells. IL-17RB immunofluorescence staining (green) of naive murine PSCs compared to PSCs treated with CM from Panc02 tumor cells for 48 h (upper row). A human primary PSC preparation increases IL-17RB expression when cultured for 48 h with conditioned CM from L3.6pl tumor cells (lower row). Scale bar = 100 μm; (**D**) Stromal cells express IL-17B. IL-17B expression by cell type in human PDAC on single-cell RNA sequencing of a public database (Genome Sequence Archive PRJCA001063). Each dot represents a single cell expressing IL-17B. PSCs are contained in the fibroblast compartment in healthy tissue and differentiate into CAFs in the tumor environment. In the first two columns, fibroblasts from healthy tissue were compared with CAFs. Fibroblast (*n* = 5540) vs. CAF (*n* = 5769; *p* = 0.004 *t*-test), tumor cells in full epithelial-to-mesenchymal transition (fEMT, *n* = 1345) vs. tumor cells in partial epithelial-to-mesenchymal transition (pEMT, *n* = 684; *p* = 0.0271, *t*-test); (**E**) IL-17B expression in tumor and paired non-malignant pancreatic tissues from publicly available TCGA and GTEx data sets analyzed with the GEPIA2 portal; (**F**) Increased IL-17B expression in pancreatic cancer tissue compared to adjacent non-tumor tissue (normal) tissue. ELISA of specimens of in-house PDAC patients (*n* = 7; *p* = 0.0263; Wilcoxon matched-pairs test); (**G**) Western blot on human primary PSC preparations of IL-17RB and α-SMA after stimulation with 200 respective 400 ng/mL rhIL-17B; (**H**) Extracellular vesicles (EVs) from human pancreatic cancer cell lines induced the expression of IL-17RB in PSCs. IL-17RB Western blot of PSCs treated with EVs isolated from two human pancreatic cancer cell lines (L3.6pl and Panc-1). The Data represent the typical images of three independent experiments; (**I**) CD63 and CD81 are specific markers for extracellular vesicles. Data represent the typical images of three independent experiments; (**J**) ELISA of exosome IL-17B expression normalized to serum. * *p* < 0.05, ** *p* < 0.01.

### 3.2. Increased Expression of the IL-17B Receptor Rewires PSCs Metabolism via Reducing Mitochondria Fission

To investigate the impact of IL-17RB upregulation on PSC activation, we stably transfected human PSCs to construct IL-17RB overexpressing and knockdown PSCs (IL-17RB OE and IL-17RB KD). IL-17RB OE PSCs are a model for tumor cell-stimulated PSCs in the following experiments. Patients undergoing pancreatic resection for chronic pancreatitis were donors for the original PSC cell line (RLT-PSC), a kind gift of Rainer Heuchel. IL-17RB-overexpressing RLT-PSCs showed stable IL-17RB expression (Figure 2A) and tended to increase expression of the PSC activation marker α-SMA (Figure 2B), as well as increased proliferation compared with control PSCs (Figure 2C). Moreover, IL-17RB OE PSCs displayed decreased expression (see Figure 2D) and secretion of IL-6 (see Figure 2E). IL-17RB KD PSCs showed respective opposite expression patterns. These data suggest that IL-17RB upregulation activates PSC.

We found a trend towards a decreased expression of p62 in IL-17RB OE PSCs, a marker of autophagy. There was no change in Beclin1-expression, indicating uncompromised macro autophagy. We observed trends towards decreased Parkin (PRK8) expression, suggesting reduced mitophagy in IL-17RB OE PSCs (see Figure 2F). Mitochondrial fission factor (MFF) phosphorylation (P-MFF) promotes mitochondrial fission and mitophagy. We found a trend towards less P-MFF expression in IL-17RB OE PSCs, indicating reduced fission and mitochondrial fragmentation. Consecutively dynamin-related protein 1 (Drp1) was released from the outer mitochondrial membrane and accumulated in the cytosol. Mitofusin-2 expression (MFN2) did not change, indicating unaltered mitochondrial fusion. Thus, decreased fission hints at mitochondrial accumulation and life extension. Again, IL-17RB KD PSCs tended to show the reverse effects (see Figure 2G). The morphology of the mitochondria of IL-17RB OE PSC differed strikingly from that of VEC OE PSC. The IL-17RB OE PSCs showed elongated mitochondria with tubular cristae (white arrows, see Figure 2H).

The reduced mitophagy may indicate an enlarged mitochondrial pool [22]. Thus, we asked if mitochondria of IL17RB OE PSCs change functionally towards increased mitochondrial activity and energy metabolism. We found a significantly higher oxygen consumption rate (OCR) in IL-17RB OE PSCs than cells transfected with an empty vector (VEC OE, Figure 3A). Through assessment of the extracellular acidification, we determined a lower glycolytic Proton Efflux Rate (glycoPER) in IL-17RB OE PSCs (see Figure 3B). The glycolytic Proton Efflux Rate correlates with lactate accumulation over time, and thus with glycolysis. These experiments showed that IL-17RB increased oxidative phosphorylation and decreased glycolysis in PSCs. IL-17RB knockdown PSCs (IL-17RB KD) showed decreased oxidative phosphorylation and no changes in glycolysis compared to control cells (VEC KD, see Figure 3A,B). Taken together, overexpression of IL-17RB stimulates mitochondrial respiration in PSCs.

### 3.3. IL-17RB Overexpressing PSCs Support Pancreatic Cancer Growth and Increase Oxidative Phosphorylation

Next, we validated PSC-induced tumor proliferation in vivo using a mouse xenograft model. We injected human MIA PaCa-2 pancreatic carcinoma cells with control IL-17RB VEC OE PSCs respective IL-17RB OE PSCs into immunocompromised mice (NSG). MiaPaCa-2 cells were chosen because they were already used together with PSCs in the literature [7]. Co-injection of PSCs and Mia PaCa-2 cells resulted in tumor formation with a stromal component (Figure 4A). Tumor size and weight increased significantly when tumor cells were injected together with control IL-17RB VEC OE PSCs, reflecting the cooperation of the tumor with stroma. Compared with control PSCs, tumor size and weight again significantly increased when co-injected with IL-17RB OE PSCs (Figure 4B). Co-injected control IL-17RB VEC OE PSC also accelerated tumor growth in animals over time. IL-17RB OE PSCs again significantly accelerated tumor growth compared with control PSCs (Figure 4C).

This result is consistent with experiments showing that conditioned medium (CM) of IL-17RB OE PSCs (CM OE) increases pancreatic cancer cell growth, whereas CM of control PSCs (CM VEC OE) does not. CM from IL-17RB KD PSCs (CM KD) increased the growth of pancreatic cancer cells to the same extent as CM from IL-17RB VEC KD PSCs (CM VEC KD). We attribute the increase by CM VEC KD compared with CM VEC OE to the different vectors used to induce overexpression or knockdown. Only the CM VEC OE group can be compared with the CM OE group and the CM VEC KD group with the CM KD group. Comparison of the CM VEC OE and CM VEC KD groups is not reasonable (Figure 4D). CM of IL-17RB OE PSCs induced CXCR4 expression (Figure 4E), contributing to tumor growth, invasion, and metastasis [23]. Molecularly Signal Transducer and Activator Of Transcription 3 (STAT3)-phosphorylation and hexokinase 2-expression were reduced in human cancer cells treated with CM from IL-17RB OE PSCs (Figure 4F). Hexokinase 2 catalyzes the first step of glycolysis. Furthermore, we found that exosomes from IL-17RB OE PSCs (exosome OE) increased the sphere formation rate compared with exosomes from control IL-17RB VEC OE PSCs (Figure 4G). Exosomes from IL-17RB OE PSCs increased the sphere size to the same extent as exosomes from control PSCs (exosome VEC OE, Figure 4H).

Based on the IL-17RB induced metabolic changes in PSCs, we hypothesized that PSCs affect the metabolism of pancreatic cancer cells. To test how soluble factors of PSCs influence the energy turnover of tumor cells, we cultured PDAC cells with the conditioned medium (CM) of PSCs. We measured the oxygen consumption rate (OCR) of human Panc-1 tumor cells cultured alone or with CM from IL-17RB OE (and from the respective VEC OE control), as well as CM from IL-17RB KD PSCs (and VEC KD control). Adding control CM (VEC OE) from PSCs transfected with an empty vector increased the OCR slightly compared with naive tumor cells. PDAC cells treated with CM from IL-17RB OE PSCs showed a trend towards increased OCR compared with control CM and a significant increase of the OCR compared with naive tumor cells (Figure 5A). Adding CM of control PSCs, transfected with an empty vector (VEC KD), produced a significantly increased OCR than naive Panc-1 cells (Figure 5A). CM of IL-17RB KD PSCs tended to decrease the OCR in Panc-1 tumor cells compared with control PSCs (Figure 5A). Thus, IL-17RB overexpression in PSC might stimulate oxidative phosphorylation in PDAC cells, while IL-17RB knockdown PSCs could attenuate oxidative phosphorylation.

We assessed the glycolytic proton efflux rate (glycoPER), which measures glycolysis and extracellular lactate production. Adding CM from control PSCs (CM VEC OE) to Panc-1 cells reduced the basal glycolysis. The reduction became significant when adding CM from IL-17RB OE PSCs (CM OE, Figure 5B). The basal glycolysis did not change when adding CM from IL-17RB KD PSCs (CM KD) and control PSCs (CM VEC KD). However, CM from control PSCs (VEC KD CM) decreased compensatory glycolysis. The reduction was significant when adding CM from IL-17RB KD PSCs (KD CM, Figure 5B). Trends toward similar results depicted analogous experiments with a second PDAC cell line (Mia PaCa-2, Appendix A).

### 3.4. Mitochondria Express IL-17RB

Because not IL-17B but IL-17B-carrying exosomes upregulated IL-17RB, we asked whether PSCs take up IL-17B intracellularly and what effects IL-17B could exert there. Coexpression with ATP5β, a subunit of the mitochondrial ATP synthase, reveals the localization of IL-17RB on the mitochondria of human and murine PSCs and PDAC cells using confocal immunofluorescence microscopy (Figure 6A). Through analysis of the individual cell compartments, we confirmed mitochondrial IL-17RB expression with Western blot in PSCs and pancreatic cancer cells (Figure 6B). In IL-17RB OE PSC, the majority of IL-17B receptors are located on the mitochondria. We also detected IL-17B in the mitochondrial compartment of PSCs (Figure 6C) so that a mitochondrial IL-17B/IL-17RB signaling is in principle possible.

## 4. Discussion

Our study reveals an IL-17B/IL-17RB mediated bidirectional crosstalk between PDAC cells and PSCs. Tumor cell-derived IL-17B carrying extracellular vesicles induces the expression of the IL-17B receptor on PSCs. IL-17RB overexpression results in downregulation of proteins controlling mitochondrial fission and mitophagy, increasing mitochondrial activity. Mitochondrial-derived ATP production increases in PSCs through enhanced oxidative phosphorylation. Through soluble factors, including IL-6, PSCs induce a feedback response in tumor cells which are also metabolically activated. The ATP production rate in tumor cells increases by enhanced mitochondrial respiration, while glycolysis decreases. Overexpression of IL-17RB, as well as stimulation with IL-17B, decreases PSCs IL-6 secretion. Tumor cells reduce STAT3 signaling [24], downregulate hexokinase 2, and reduce glycolysis. Thus, IL-17B/IL-17RB signaling establishes a feedback loop between tumor cells and PSCs. Tumor and stroma activate each other to increase overall energy production leading to tumor progression (Figure 6D).

We can furthermore show for the first time that mitochondria express the IL-17B receptor. Tumor cell-derived extracellular vesicles transport IL-17B into the cytosol of PSCs. The mitochondrial effect seen in our experiments may indicate mitochondrial IL-17B/IL-17RB signaling. A limitation of the study is that signal transduction within mitochondria was not detected. Therefore, there is only indirect evidence of intracellular cytokine signaling. Further experiments are needed to demonstrate a mitochondrial metabolic response to intracellular cytokines.

Recent studies have shown metabolic interactions between stroma and tumor cells in various types of cancer [25,26,27]. These interactions can promote or inhibit tumor growth. Complete deletion of the stroma leads to pancreatic cancer progression, demonstrating that stromal compartments can also have anti-tumoral functions [16]. Selective depletion of PSCs and CAFs based on their α-SMA expression reduces survival in tumor-bearing mice [15]. Contrarily, Sousa et al. demonstrated that PSCs promote tumor growth through autophagy and consecutive alanine secretion [7]. Also, Shi et al. identified leukemia inhibitory factor (LIF) as an essential PSC-derived paracrine factor that stimulates cancer cell growth [28]. Our study demonstrates the tumor-promoting effect of PSCs via IL-17B/IL-17RB signaling. The idea that a single cell type can either have tumor-promoting or tumor-inhibiting properties depending on its activation state balances these observations. Öhlund et al. provide a clue to this explanation, demonstrating opposing effects on tumor development by distinct PSC-derived CAF populations [5]. However, it is also conceivable that a single cell type such as PSCs has simultaneous tumor-promoting and inhibitory effects. Depending on the microenvironment, one of these effects predominates. Targeted inhibition of the protumorigenic properties without complete PSC elimination could preserve the protective PSC effect. This approach could make the above-discussed observations therapeutically applicable. A recent example of this strategy has shown that targeted pharmacological LIF blockade slows tumor progression in a mouse model [29]. It can be speculated that targeting the IL-17B receptor on PSCs could inhibit tumor growth, too.

The role of glycolysis versus oxidative phosphorylation in tumor cells is under debate. Central to the form of the tumor cell’s energy production is the availability of oxygen. Because of the hypoxic conditions in pancreatic cancer, tumor cells primarily rely on glycolysis. The “Warburg effect” describes that cancer cells even use ineffective glycolysis when oxygen is available. However, cancer cells can also switch to oxidative phosphorylation to adapt to environmental changes [30]. The other key factor in energy production is nutrient availability. The stroma can form symbiotic metabolic interactions with tumor cells to compensate for nutrient deficiencies and regulate anaerobic glycolysis as well as oxidative phosphorylation. PSCs, for example, can provide alanine through autophagy to maintain pancreatic cancer cells’ high-energy requirements [7]. Alanine induces oxidative phosphorylation in cancer cells, enabling survival and cell growth in a highly nutrient-poor microenvironment [7]. It is unclear whether this process is sustainable in the long term. Prolonged autophagy could degrade PSCs that lose their nutrient reserves.

PSCs do not provide nutrients via IL-17B/IL-17RB signaling. Rather, PSCs upregulate oxidative phosphorylation and inhibit glycolysis in tumor cells. In our experiments, sufficient oxygen and optimal nutrient conditions were always present. The question is, under what conditions do tumor cells acquire an advantage when PSC “switch” them from anaerobic glycolysis to oxidative phosphorylation. Near blood vessels, well-oxygenated cancer cells might benefit from a metabolic switch towards oxidative phosphorylation. A possible scenario where the IL-17B/IL-17RB mediated metabolic shift is highly relevant is liver metastasis. In the liver, tumor cells find optimal nutrient and oxygen conditions. It has been shown that hepatic Kupffer cells take up PDAC-derived exosomes and induce premetastatic niches in the liver to increase the metastatic burden [31]. It remains to be shown if hepatic stellate cells (HSCs), the counterpart of PSCs in the liver, can also take up exosomes. Possibly, HSCs promote liver metastasis via exosome-mediated IL-17B/IL-17RB signaling by increasing oxidative phosphorylation in metastatic tumor cells.

## 5. Conclusions

Our results suggest a pro-tumorigenic effect of stroma-to-tumor IL-17B/IL-17RB signaling and that specific inhibition of the IL-17B receptor could be an effective therapy to suppress the metabolic tumor-stroma cooperation.

## Figures and Tables

**Figure 2 cancers-13-05338-f002:**
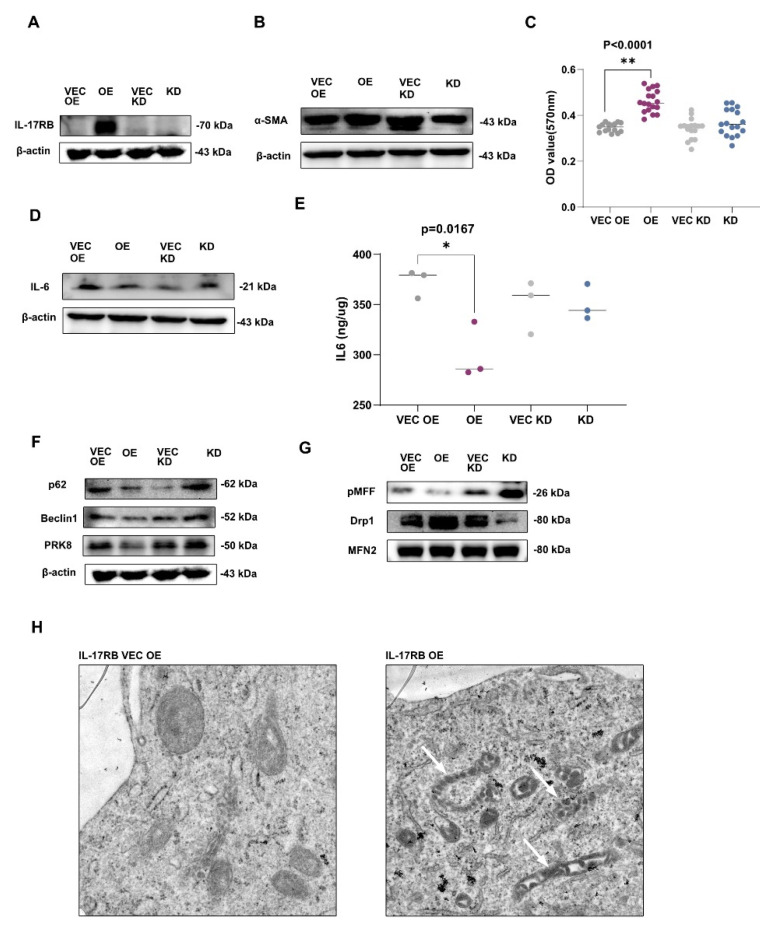
IL-17RB alters mitochondrial turnover and IL-6 secretion. (**A**) IL-17RB OE PSCs stably express IL-17RB without activation. IL-17RB KD PSCs do not express IL-17RB; (**B**) IL-17RB OE PSCs increase α-SMA. IL-17RB and α-SMA Western blot of IL-17RB OE and IL-17KD cells with corresponding control of IL-17RB VEC OE and IL-17RB VEC KD cells; (**C**) IL-17RB overexpression increases PSCs proliferation. MTT assay comparing IL-17RB OE, IL-17RB KD PSCs with respective controls (IL-17RB VEC OE and VEC KD, *p* < 0.0001, Mann–Whitney U test). Data represent the mean of three independent experiments conducted in sextuplicate; (**D**) IL-17RB overexpression decreases IL-6 expression. IL-6 Western blot of IL-17RB OE and IL-17KD PSCs with corresponding controls; (**E**) IL-17RB overexpression decreases IL-6 secretion, while IL-17RB knockdown induces no changes. ELISA of cell culture supernatant (*n* = 3, OE vs. VEC OE, *p* = 0.0167, Mann-Whitney U test); (**F**) Auto-/mitophagy Western blot of IL-17RB OE and IL-17KD PSCs with corresponding controls. p62—Autophagy receptor p62, PRK8—Ubiquitin E3 ligase Parkin; (**G**) IL-17RB overexpression reduces mitochondrial fission and fragmentation. Western blot of IL-17RB OE and IL-17RB KD PSCs with corresponding controls. pMMF—phospho-mitochondrial fission factor, DRP1—phospho-Dynamin related protein 1; (**H**) IL-17RB overexpression induces elongated mitochondria with tubular cristae (white arrows). Electron microscope images of IL-17RB OE PSCs compared to IL-17RB VEC OE controls. Scale bar = 250 nm. * *p* < 0.05, ** *p* < 0.01.

**Figure 3 cancers-13-05338-f003:**
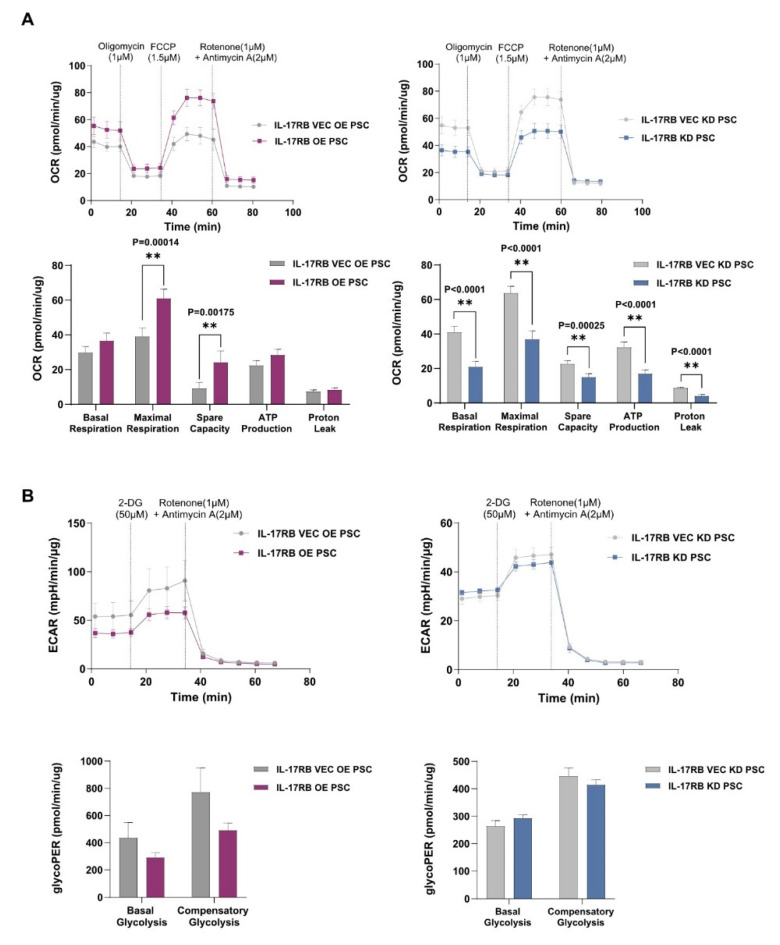
IL-17RB overexpression induces mitochondrial OXPHOS in PSCs (**A**) IL-17RB overexpression increases and IL-17RB knockdown decreases mitochondrial respiration in PSCs. Oxygen consumption rate (OCR) was measured with the Seahorse mitostress test. Seahorse profiles (upper panel) and statistical analysis (unpaired *t*-test, lower panel). Data represent the mean (±SD) of three independent experiments conducted in quintuplicate; (**B**) IL-17RB overexpression decreases glycolysis in PSCs while IL-17RB knockdown induces no changes. Statistical significance was not reached in either group. The glycolytic proton efflux rate (glycoPER) correlates with lactate accumulation over time, showing glycolysis: seahorse profiles (upper panel) and statistical analysis (unpaired *t*-test, lower panel). Data represent the mean (±SD) of three independent experiments conducted in quintuplicate. ** *p* < 0.01.

**Figure 4 cancers-13-05338-f004:**
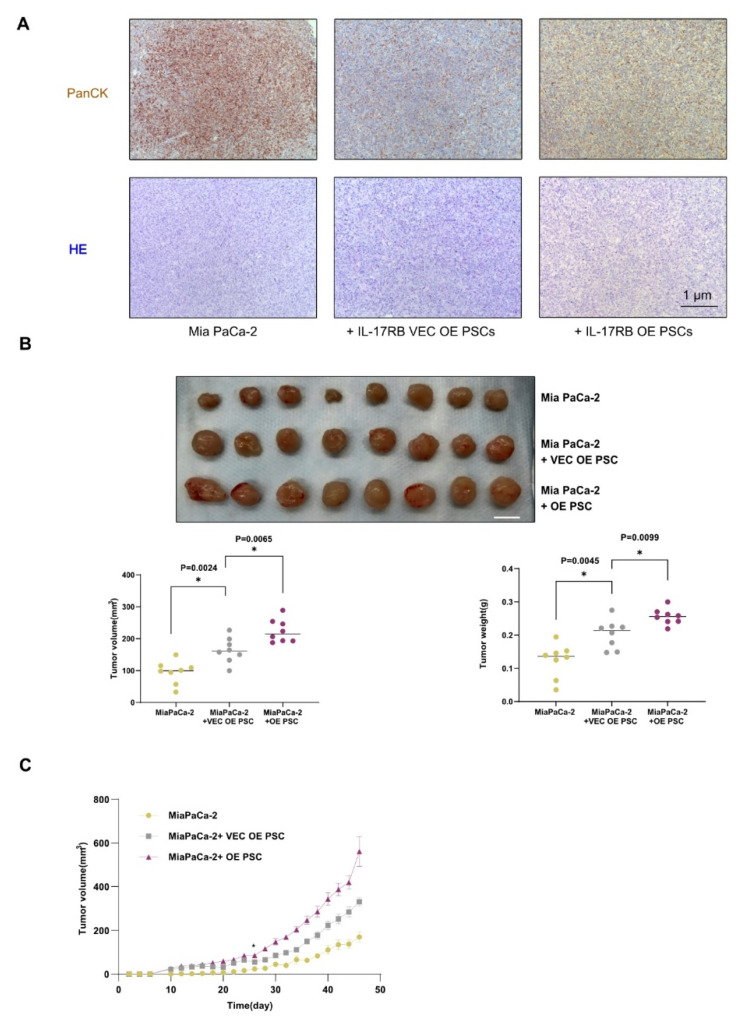
IL-17RB expressing PSCs promote tumor growth (**A**) Immunohistochemistry staining of pan-cytokeratin (brown, upper panel) and hematoxylin and eosin (H&E) of tumors developing after co-injection of PSCs and Mia PaCa-2 cells; (**B**) Control IL-17RB VEC OE PSCs and IL-17RB OE PSCs accelerate the growth of MIA PaCa-2 tumors. Image of tumors harvested at day 46 from NSG mice after co-injecting MIA PaCa-2 cells and IL-17RB OE PSCs (OE, *n* = 8) respective control PSCs (VEC OE, *n* = 8, upper panel). Statistics of tumor volume (*n* = 8; Mann-Whitney U test, *p* = 0.0024 MIA PaCa-2 vs. VEC OE, *p* = 0.0065 VEC OE vs. OE, lower left panel) and weight (*n* = 8; Mann-Whitney U test, *p* = 0.0045 MIA PaCa-2 vs. VEC OE, *p* = 0.0099 VEC OE vs. OE) at day 46 after injection (lower right panel). Scale bars = 1 cm; (**C**) Tumor growth curve of animals injected with MIA PaCa2-cells or tumor cells with OE PSCs or VEC OE PSCs (all groups, *n* = 8). VEC OE PSC and OE PSC curve significantly diverged from MIA PaCa-2 tumors at day 10, comparing individual time points with *t*-tests. The OE PSC curve significantly diverged from the VEC OE PSC curve on day 26; (**D**) The conditioned medium (CM) of IL-17RB overexpressing PSCs (CM OE) increases the growth of L3.6pl pancreatic cancer cells. CM of control PSCs transfected with the corresponding empty vector (CM VEC OE) did not. The conditioned medium of IL-17RB knockdown PSCs (CM KD) increases tumor cell growth to the same extent as corresponding CM from control PSCs (CM VEC KD). MTT cell assay to assess cell proliferation; (**E**) Conditioned medium from IL-17RB OE PSCs but not from IL-17RB KD PSCs increases CXCR4 expression in L3.6pl pancreatic cancer cells. Flow cytometry analysis of FSC-A/CXCR4 positive cells. Data represent the mean (±SD) of three independent experiments performed in triplicate; (**F**) IL-17RB overexpression inhibits the IL-6 signaling pathway. Western blot of MIAPaCa-2 cells treated with the conditioned medium of IL-17RB OE and IL-17RB KD cells with corresponding controls. STAT3—Signal transducer and activator of transcription 3, pSTAT3 phospho-STAT3. Data represent the mean (±SD) of three independent experiments; (**G**) IL-17RB OE PSC-derived exosomes increased the formation rate of L3.6pl and HPAF-II (*n* = 5) tumorspheres compared with exosomes from control PSCs. Statistical analysis of a tumorsphere formation assay (Mann–Whitney U test; *p* = 0.0079, L3.6pl vs. L3.6pl + exo OE; *p* = 0.0079, L3.6pl + exo OE vs. L3.6pl + exo VEC OE; *p* = 0.3413, L3.6pl vs. L3.6pl + exo VEC OE; *p* = 0.0079, HPAF-II vs. HPAF-II + exo OE; *p* = 0.0114, HPAF-II + exo OE vs. HPAF-II + exo VEC OE; *p* = 0.0079, HPAF-II vs. HPAF-II + exo VEC OE). Data represent the mean (±SD) of three independent experiments conducted in sextuplicate. (**H**) Exosomes from IL-17RB OE PSCs (exosome OE) increase the size of L3.6pl and HPAH-II tumor spheres to the same extent as exosomes from control PSCs (exosome VEC OE). Representative images of a tumorsphere formation assay. Scale bars = 50 μm. * *p* < 0.05.

**Figure 5 cancers-13-05338-f005:**
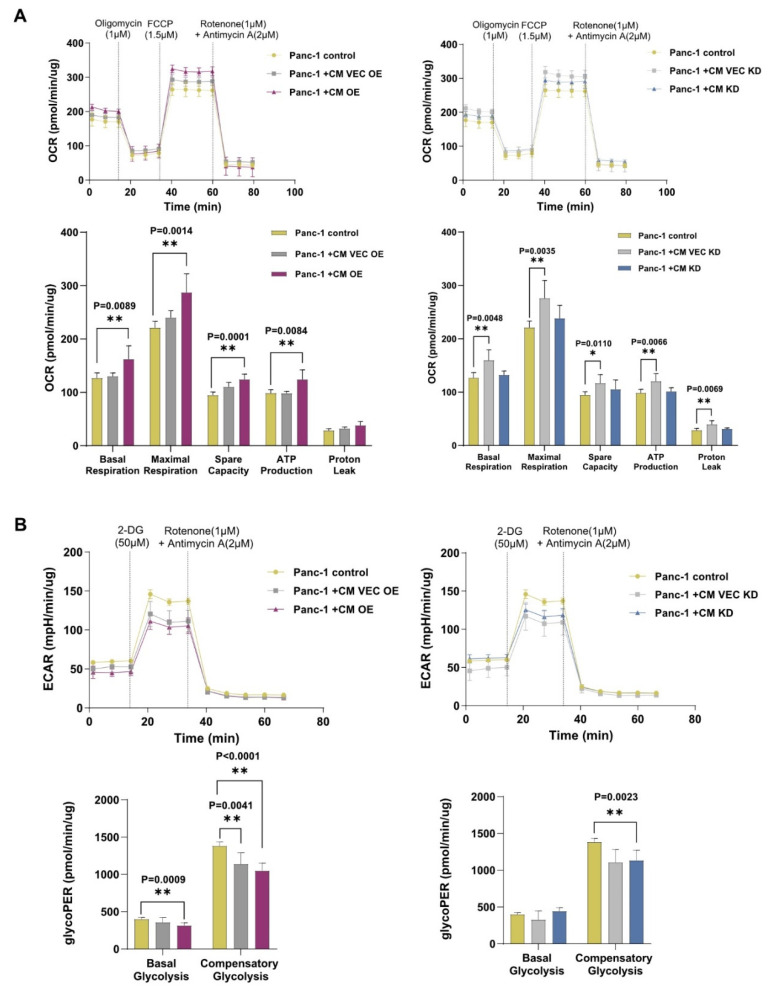
IL-17RB overexpressing PSCs increase mitochondrial respiration and decrease glycolysis (**A**) Conditioned medium from IL-17RB OE PSCs (CM OE) but not from IL-17RB KD (CM KD) increases mitochondrial respiration in Panc-1 tumor cells. Oxygen consumption rate (OCR) was measured with the Seahorse mito stress test. Seahorse profiles (upper panel) and statistical analysis (unpaired *t*-test, lower panel). Data represent the mean (±SD) of three independent experiments conducted in sextuplicate; (**B**) Conditioned medium from IL-17RB OE PSCs (CM OE, *n* = 6) decreases glycolysis in Panc-1 tumor cells. The glycolytic proton efflux rate (glycoPER) correlates with lactate accumulation over time, showing glycolysis. Seahorse profiles (up) and statistical analysis (unpaired *t*-test, down). Data represent the mean (±SD) of three independent experiments conducted in quintuplicate. * *p* < 0.05, ** *p* < 0.01.

**Figure 6 cancers-13-05338-f006:**
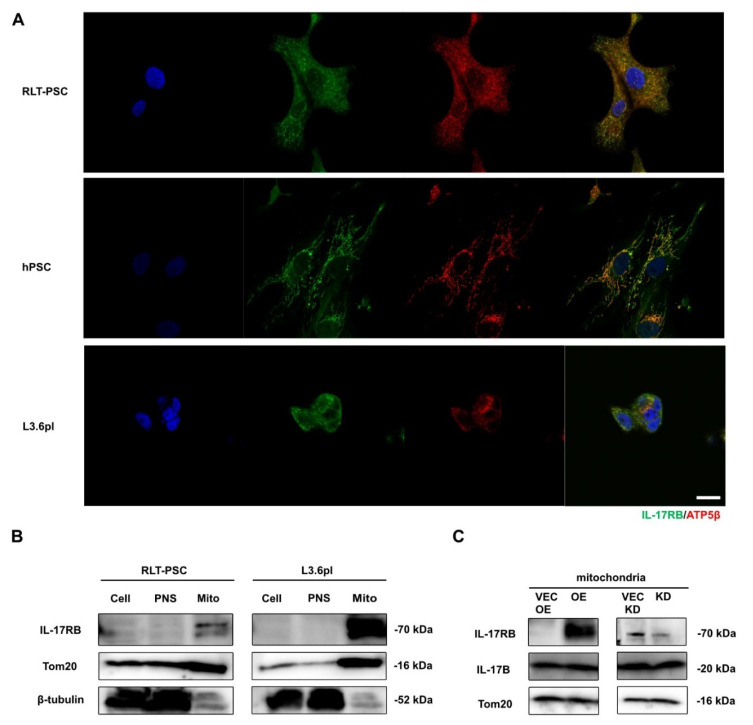
Subcellular location of IL-17RB in PDAC tumor and PSC cells (**A**) IL-17RB is localized on the mitochondria of PSCs (RLT-PSC cell line, human primary PSC preparations) and L3.6pl tumor cells. Representative confocal immunofluorescence microscopy images of ATP5β/IL-17RB double staining (ATP5β—subunit of the mitochondrial ATP synthase). Scale bars = 20 μm; (**B**) Western blot of RLT-PSCs and L3.6pl tumor cells after separating the cell compartments (Cell—whole cells, PNS—perinuclear space, Mito—mitochondrial space); (**C**) Western blot of the mitochondrial compartment of IL-17RB OE and IL-17RB KD PSCs with corresponding TOM20 control (translocase of outer membrane). Data represent typical images of three independent experiments; (**D**) Summary. Tumor and stroma activate each other to increase mitochondrial respiration leading to tumor progression. Tumor cells release extracellular vesicles (EVs) carrying IL-17B. PSCs induce IL-17RB expression leading to increased mitochondrial respiration. PSCs reduce mitochondrial fission and mitophagy. Upon IL-17B stimulation, PSCs decrease IL-6 secretion. Tumor cells reduce STAT3 signaling. The tumor cells downregulate hexokinase 2 and decreasing glycolysis. The tumor cells are metabolically activated with increased mitochondrial respiration. Thus, IL-17B/IL-17RB signaling establishes a feedback loop between tumor cells and PSCs.

## Data Availability

Publicly available datasets were analyzed in this study. This data can be found here: https://bioturing.com/, accessed on 23 Octorber 2021 and here http://gepia2.cancer-pku.cn, accessed on 23 Octorber 2021 (see Methods sections for details). The data presented in this study is contained within the article and Appendix A.

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
