# Peer review of "IL-17B/RB Activation in Pancreatic Stellate Cells Promotes Pancreatic Cancer Metabolism and Growth"

_cancers, 2021, doi:10.3390/cancers13215338_

Round 1

Reviewer 1 Report

In the present manuscript entitled: IL-17B/RB Activation in Pancreatic Stellate Cells Promotes Pancreatic Cancer Metabolism and Growth, Li and colleagues describe experiments to study the role of interleukin 17B (IL-17B) in the interaction between pancreatic stellate cells (PSCs) and pancreatic adenocarcinoma cells and the progression of pancreatic cancer.

The authors show evidence that IL-17B can be carried by extracellular vesicles and supply data that its stimulation of PSCs with IL-17B modulates the oxidative phosphorylation of mitochondria and decreases mitochondrial turnover. The results support a positive influence of IL-17B on tumor growth.

The study contains a highly interesting and relevant topic, the presented data are of significant interest to readers from the oncological field. The study seems to be well designed and the experiments are fairly arranged and conducted.

Unfortunately, the manuscript itself seems somewhat preliminary and does not justify publication in its present form.

My criticism can be summerised as follows:

The whole manuscript has to be carefully revised and rewritten.

- Even when the authors are familiar with (standard) abbreviation such as IL-17B, IL-17RB, OXPHOS, CAFs etc., all abbreviations should be explained at their first use.

- The introduction is very short and should contain some information about mitochondrial metabolism or activity which is a main topic in the Results

- Material + Methods contain chapter 2.10. Pancreatic stellate cell and cancer-associated fibroblast isolation and immortalization. But there is no word about immortalization in the text.

- The description of the cell culture is not sufficient. Which cell lines have been used in the study?

- What is about the first sentences in Results and Discussion? Are they relicts of a master copy? These sentences are unnecessary.

- The description in the results is most often not consequently structured and scientifically not sound: (Some examples)

The experimental design is not always reproducible: Why did the authors co-culture murine and human PSCs (line 254)?

How did the authors produce conditioned medium from pancreatic cancers (line 259)? I suggest they mean the MiaPaCa cell line. The procedure is not mentioned in M+M. Please be more precisely, a tumour cell line is not a tumor and only a poor representative of the appropriated tumour type. As a conclusion (line 220) it should be stated that MiaPaCa cells are capable to induce IL-17RB, the ability of pancreatic cancers or more general tumor cells was not analysed in the experiments.

It has been widely accepted that the use of two different cell lines is necessary to validate the observed effects.

The authors should include data how they characterised the quality and purity of the isolated primary PSCs and cancer associated fibroblasts (CAFs).

The EM pictures in Fig. 2F are very small and the described details are hardly recognisable.

Why did the authors used exosomes in contrast to conditioned medium to modulate the size of spheres?

- Some conclusions drown by the authors in the Results chapter seem hardly supported by the presented data.

Where did the authors analyse the pluripotency of PSCs as stated in line 302/303?

Many conclusions are presented as causative explanations but most are not supported by the presented data and are correlative events at its best.

And many more points during the Results chapter

Author Response

Reviewer 1

The whole manuscript has to be carefully revised and rewritten.

We went through the manuscript and carefully improved it. Please refer to the following statements for the individual changes.

- Even when the authors are familiar with (standard) abbreviation such as IL-17B, IL-17RB, OXPHOS, CAFs etc., all abbreviations should be explained at their first use.

We have carefully reviewed the manuscript and introduced all abbreviations at the first mention.

- The introduction is very short and should contain some information about mitochondrial metabolism or activity which is a main topic in the Results

We have added a paragraph to introduce mitochondria as follows:

We hypothesize that a tumor-promoting interaction of PSCs and tumor cells via IL-17B/IL-17RB may affect energy metabolism. Increased energy production would lead to tumor growth and would have to be mediated by mitochondria. Mitochondria are organelles that generate energy in the form of adenosine triphosphate (ATP) through oxidative phosphorylation (OXPHOS). Mitochondria undergo fission and fusion processes in response to changes in energy demand and cellular stress. These processes may also play a role in tumorigenesis. The three main steps of mitochondrial fusion are tethering of two mitochondria, docking of the membranes, and fusion of the outer mitochondrial membranes (OMM). Fusion of OMM is induced by two GTPases, mitofusin1 (MFN1) and mitofusin2 (MFN2)[Science, 2004. 305(5685): p. 858-62.]. The GTPase dynamin-related protein 1 (Drp1) plays an essential role in mitochondrial fission [Essays Biochem, 2018. 62(3): p. 341-360.; Mol Cells, 2018. 41(1): p. 18-26.]. Among the organelles that can be degraded by autophagy are mitochondria. The selective degradation of mitochondria by autophagy is referred to as mitophagy [Dev Cell, 2017. 41(1): p. 10-22.]. It has been demonstrated that extracellular lactate can promote tumor mitochondrial energy production. One of the theories describing the metabolic coupling between the tumor cell and cancer-associated fibroblasts (CAFs) is the reverse Warburg effect. Here, tumor-induced glycolysis in the CAF supplies adjacent cancer cells with lactate, inducing the tricarboxylic acid (TCA) cycle for ATP generation. This increases tumor proliferation and decreases cell death.

- Material + Methods contain chapter 2.10. Pancreatic stellate cell and cancer-associated fibroblast isolation and immortalization. But there is no word about immortalization in the text.

For stimulations of PSCs with IL-17B, higher cell numbers were needed (Figure 7). SV40 large T antigen encoded by a plasmid were transfected into primary PSCs using a lentivirus to immortalize cells[Jesnowski, R., et al.Lab Invest, 2005. 85(10): p. 1276-91]. Immortalization allowed the culture of large cell numbers. PSCs were used within 10 passages. We added to the M&M as well to the main text to clarify this.

- The description of the cell culture is not sufficient. Which cell lines have been used in the study?

We rewrote the description of the cell cultures in the materials and methods as follows (lines 248-255):

The human pancreatic cancer cell lines MiaPaCa-2, Panc-1, and HPAF-II were purchased from the American Type Culture Collection (ATCC). The immortalized pancreatic stellate cell line RLT-PSC was kindly provided by Reiner Heuchel and Martin Löhr (Karolinska-Institute, Sweden) and has been previously described [15]. The human pancreatic cancer cell line L3.6pl was kindly provided by Dr. G.E. Gallick (University of Texas MD Anderson Cancer Center, Houston, TX) [16].

- What is about the first sentences in Results and Discussion? Are they relicts of a master copy? These sentences are unnecessary.

We deleted the first two sentences of the results and discussion section to make the manuscript clearer.

- The description in the results is most often not consequently structured and scientifically not sound: (Some examples)

The experimental design is not always reproducible: Why did the authors co-culture murine and human PSCs (line 254)?

We did not co-culture murine PSCs with human PSCs but murine PSCs with murine tumor cells (Panc02) and human PSCs with human tumor cells (L3.6pl). We think that the results are more robust when reproduced in two species. To make this approach clearer, we changed the text and legend of figure 1B.

How did the authors produce conditioned medium from pancreatic cancers (line 259)? I suggest they mean the MiaPaCa cell line. The procedure is not mentioned in M+M.

We prepared the conditioned medium (CM) from the murine PDAC cell line Panc02 and the human PDAC cell line L3.6pl. For this, we cultured the tumor cells for 48 hours to a confluency of 70-90%. Then, we centrifuged the CM from the cell culture. These experiments were done in analogy to those in figure 1B. We changed the ordering of the figures, the figure legend, and the result section to clarify this. We added a paragraph to the M&M section to describe how we prepared the conditioned medium.

Please be more precisely, a tumour cell line is not a tumor and only a poor representative of the appropriated tumour type. As a conclusion (line 220) it should be stated that MiaPaCa cells are capable to induce IL-17RB, the ability of pancreatic cancers or more general tumor cells was not analysed in the experiments.

The reviewer is correct in his statement. We changed the conclusion and made it clear that only specific pancreatic cancer cells induce IL-17RB.

It has been widely accepted that the use of two different cell lines is necessary to validate the observed effects.

We demonstrated the results discussed above using tumor cell lines from mice and humans. We believe that the reproducibility of the experiments in two species sufficiently validates our observations.

We repeated the key experiments from figure 5 with the MiaPaCa-2 cell line and created a supplementary figure 1.

The authors should include data how they characterised the quality and purity of the isolated primary PSCs and cancer associated fibroblasts (CAFs).

We used vimentin and a-SMA immunofluorescence staining to determine purity and control quality. A 95% positivity for vimentin and typical morphology indicated an acceptable isolation of PSCs. The same method was used to isolate CAFs. CAFs overgrow other cell types during the cell culture.

The EM pictures in Fig. 2F are very small and the described details are hardly recognisable.

We increased the size of the EM images and marked the changes in IL-17RB OE PSCs with arrows.

Why did the authors used exosomes in contrast to conditioned medium to modulate the size of spheres?

In our hands, no spheres formed at all when conditioned medium (CM) was added. In our experience, the use of CM leads to a highly fluctuating concentration of secreted proteins when CM is added several times in long-term culture of 14 days in the sphere formation assay. We used exosomes instead of CM because exosomes secrete proteins at constant concentrations.

- Some conclusions drown by the authors in the Results chapter seem hardly supported by the presented data.

We took the reviewer's advice very seriously and worked through the conclusions one by one and corrected them.

We have removed the statements on the expression of CAV-1 and CCL-5 and believe that this does not change the overall message of the paper. However, we see a difference in IL-6 expression and have supported this with a quantification of the Western blots (see attached file “Western blot quantification.pdf”. We have prepared these graphics for the reviewer and have not included them in the manuscript in order not to exceed its scope.).

We reduced Figure 2C to show only P-MFF, Drp1, and MFN2. Here, we could draw a clear conclusion. We removed the ambiguous results. We reduced Figure 2E to show the blots founding a clear conclusion. We inserted the correct interpretation in the main text. We removed figure 2D to concentrate on the main conclusion of the manuscript. The STAT3 Western blot in Figure 4C does not match the description in the results. We have reworked this part from the results section. Correspondingly, we have adjusted the figure legends.

Where did the authors analyse the pluripotency of PSCs as stated in line 302/303?

We did not analyze pluripotency. In line 302, we changed the conclusion and deleted the concept of pluripotency.

Many conclusions are presented as causative explanations but most are not supported by the presented data and are correlative events at its best.

We went through the conclusions one by one and changed them as described above. In doing so, we have emphasized correlations and removed descriptions when presented as causalities.

Reviewer 2 Report

Dear Authors, the paper is well conceived

and presented. New researches in this field can

improve dramatically the survival of pancreatic 

cancer. Minor revisions are required in my opinion before 

publication:

  • typo error in line 241/243 and 502/505;
  • further stress the limit of the study 

Author Response

Reviewer 2

typo error in line 241/243 and 502/505;

We checked spelling throughout the document and made corrections whenever necessary. We removed the lines 241/243 and 502/505 because they were relicts of a master copy.

further stress the limit of the study

We added a paragraph about the limitations of the study, starting at line 596.

Reviewer 3 Report

The work by Li et al showed interesting results on the possible metabolic cooperation between pancreatic cancer cells and the surrounding tumor stroma. They observed that a mitochondrial IL-17B/IL-17RB signaling could mediate a bidirectional crosstalk between pancreatic ductal adenocarcinoma cells (PDAC) and pancreatic stellate cells (PSCs) and suggested a possible therapeutic benefits from the specific inhibition of IL-17RB receptor.

The overall study is interesting, however, the inconsistencies in most of the results greatly impact on the quality of work and no clear conclusions can be extracted from this research.

MAJOR POINTS

The Figure 1A is poorly made. Appropriate controls labelling tumor cells and activated PSC must be shown to properly indicate that IL17RB is coexpressed in both cell type. At least a cytokeratin staining for cancer cell and more specific markers for activated PSC (such as desmin or GFAP…) must be showed to claim that “a population of activated PSC” expresses IL17RB.

There are not any details in the Materials and Methods that explained how Figure 1B was obtained and, indeed, it is not described which proportion of PDAC and PSC cells was used to perform the experiments. Moreover, it is important to know the level of expression of IL-17RB in PDAC and PSC cells and the cell proportion in co-culture experiments to clarify if the up-regulation of IL-17RB in co-culture could be due to a possible crosstalk or simply to the higher level of expression of IL-17RB in tumor cells. 

Quality of most of the western blots in Figure 2 is very low and inconsistent. Discrepancies in the controls used (VEC OE and VEC KD) are very evident and VEC KD showed the strongest effect among the treatment compared to VEC OE. Protein levels in the controls should be comparable: this compromised all the results or conclusions made from these experiments. In addition, the effects described for a-SMA and CAV-1 for the proliferation, ACLY and ACC for fatty acid synthesis and LC3B, BNIP3 and NIX for the mitophagy in IL-17RB overexpressing PSCs (IL-17RB OE) as well as the reverse effects due to the knockdown are not supported by the figures. Since the WB experiments are done in triplicate a quantification of the bands among all the experiments, showing standard deviation and statistical significance must be provided to support the conclusion described in the manuscript especially for those protein for which the authors claimed an altered expression.

Moreover, no results were shown about the efficiency of overexpression or knockdown for IL-17RB. Please provide qPCR and/or western blot of IL-17RB expression.

One of the strongest effects that the authors described is on mitochondrial fission and mitophagy. Please provide more evidence about Drp1 release from mitochondria and accumulation in the cytoplasm in the IL-17RB OE and KD by cell fractionation or immunofluorescence with all appropriate control markers.

Also, in figure 3 there is an evident discrepancy between the vehicle only controls. Please explain why the empty vectors have such effect on OCR too as was previously mentioned above for Fig2. This issue greatly impacts on the interpretation of the results. Please show the non-infected cells too, to establish if the empty vectors impact on the cellular functions discussed in this research and repeat all the relative experiments with a more suitable vector and a non-infected control.

Multiple unrelated PDAC cell lines were used for the presented assays in Figure 4 and 5, in particular the Hexokinase/STAT3 Western blot is shown in MiaPaca2 extracts rather than in the cell lines used for sphere formation (L3.6pl and HPAF-II) or conditioned media experiments. Different cell lines imply different phenotypes. Please provide a coherent set of experiments in terms of used cell lines and explain why the authors specifically used MIA PaCa2 cells for the in vivo experiments.

In figure 4a, as for figure 1A, there is no evidence about the relative composition of xenografts: an immuno-histological assessment of PDAC and PSC would better clarify this issue.

The presented WB in Figure 4C is of very poor quality, and does not reflect the stated observation on phosphorylation/expression changes. The authors should provide more clear data to justify their claims. Also, in figure legends: there is no biological replicate nor any SD to be shown in this piece of data, please correct.

Growth rates in Fig4E seem to be definitely affected by the Empty vector (VEC KD) alone when compared to control cells. Please justify this difference or repeat the experiments with more suitable vector, as previously noted in other figures.

A simple widefield acquisition of the co-staining shown in Figure 6A is not sufficient to confirm the localization of IL-17RB in mitochondria. A high z-stack resolution by confocal microscopy is required for an accurate subcellular localization.

Figure 6B:  these western blots are of very poor quality and misleading. In the left panel, beta-tubulin and Tom20 signals do not exclude each other in the different lanes showing a very poor enrichment for the RLT-PSC. Moreover, in the right panel is not clear which cell line has been analyzed and, again, why the IL-17RB level between the VEC controls are so different between them and compared to the left panel.

MINOR POINTS

The introduction about pancreatic cancer is poorly written. It is very generic with some concepts not well explained: e.g. pancreatic cancer is not the most malignant tumor today but it sets to become the second most fatal cancer by 2030 (Latenstein, A.E.J., et al., Nationwide trends in incidence, treatment and survival of pancreatic ductal adenocarcinoma. Eur J Cancer, 2020. 125: p. 83-93). Please adjust and expand this part of the introduction.

Please provide reference for “fatty acid synthesis and mitochondrial fission maintain pluripotency”.

Please indicate clearly in the figure 2F the alterations described in the results.

Please display all barplots with the individual data points as shown in figure 6D.

Numbering of the figure panels should be coherent with the order cited in the main text.

Please explain or speculate why you see an effect on IL6 or proliferation on OE sample only.

Please check time of administration of respiratory inhibitors because they do not coincide with the changing in OCR/ECAR graphs (Fig3, Fig5B).

Figure 4F: please provide reference to the sentence “cxcr4 expression contributing to tumour growth invasion and metastasis”

In the immunofluorescence images in Fig1, the results could be clearer if each staining is shown as separate channels in addition to the merge image (as in Fig6) and if a higher magnification insert is added to the panel as for the IHC data. In addition, Figures 1 and 6 should display a clear and consistent nuclear counterstaining that for example is missing in Fig1 and inconsistent in Fig6.

Figure 7A should show the IL-17B expression of all the stromal cell clusters identified in the single-cell RNAseq data. Please provide the information of stellate cells as well.

Line 392: OSC instead of PSC

Author Response

Reviewer 3

MAJOR POINTS

The Figure 1A is poorly made. Appropriate controls labelling tumor cells and activated PSC must be shown to properly indicate that IL17RB is coexpressed in both cell type. At least a cytokeratin staining for cancer cell and more specific markers for activated PSC (such as desmin or GFAP…) must be showed to claim that “a population of activated PSC” expresses IL17RB.

In the lower immunohistochemical staining (IHC) of figure 1A, tumor cells are visible morphologically. A specialized pathologist confirmed this result. These cells are a-SMA negative and IL-17RB positive. IHC is a very basic examination, but with much and long experience. The staining clearly shows that tumor cells express IL-17RB. We have changed the text and figure legend to make this clear. We added arrows to indicate IL-17RB positive tumor cells in figure 1A.

Morphology also identifies a-SMA-positive/IL-17RB-positive stromal cells (black arrows). From Figure 1D, it is clear that murine and human primary PSC preparations express IL-17RB. The reviewer is correct that it cannot be concluded that "a population of activated PSC" expresses IL-17RB. We have changed the text to clarify that activated stromal cells express IL-17RB and removed the incorrect conclusion.

There are not any details in the Materials and Methods that explained how Figure 1B was obtained and, indeed, it is not described which proportion of PDAC and PSC cells was used to perform the experiments.

We added the experimental protocol to the Materials and Method section as follows:

For co-culture experiments (Fig. 1B), 1x105 primary PSC preparations (mPSC = murine PSCs and hPSC = human PSCs) were seeded into the lower chamber of a transwell cell culture system. 1x105 pancreatic cancer cells (L3.6pl for human culture and Panc02 for murine) were seeded into the upper chamber. Cells were cultured in DMEM/f12 with 10% FBS for 72 hours before harvesting the PSCs for Western blot analysis. We isolated primary murine PSCs (mPSC) from the pancreas of healthy C57BL/6 mice by gradient centrifugation. Primary human PSCs were obtained from pancreatic surgery specimens. We used cells of passage 3-8 for the experiments (Pancreatology. 2010;10(4):434-43.).

Moreover, it is important to know the level of expression of IL-17RB in PDAC and PSC cells and the cell proportion in co-culture experiments to clarify if the up-regulation of IL-17RB in co-culture could be due to a possible crosstalk or simply to the higher level of expression of IL-17RB in tumor cells.

We used a transwell co-culture system that physically separates the tumor cells from the PSCs. Thus, there must be crosstalk via soluble factors between PSCs and tumor cells. IL-17RB was measured exclusively on PSCs, one time in the co-culture system without and one time with tumor cells. To our knowledge, this avoids contamination by tumor cells.

Quality of most of the western blots in Figure 2 is very low and inconsistent. Discrepancies in the controls used (VEC OE and VEC KD) are very evident and VEC KD showed the strongest effect among the treatment compared to VEC OE. Protein levels in the controls should be comparable: this compromised all the results or conclusions made from these experiments.

Backbone plasmids have a fundamental impact on cell gene expression (Cold Spring Harb Protoc. 2020 Aug 3;2020(8):095513.; Genes (Basel). 2020 Mar 10;11(3):291.). Different backbones induced different tag proteins, changing the downstream protein production. In this study, different backbones were used for the overexpression (OE) and knockdown (KD) of IL-17RB for technical reasons. Thus, we explain the changes in protein expression, especially of the empty knockdown vector (VEC KD) that the reviewer correctly observed. In the experiments, only the VEC OE group can be compared with the OE group and the VEC KD group with the KD group. Comparison of the VEC OE and VEC KD groups is not reasonable because of the different backbones. We have clarified this in the Materials and Methods section and have gone through the results section again. We have improved the parts that do not take this into account.

In addition, the effects described for a-SMA and CAV-1 for the proliferation, ACLY and ACC for fatty acid synthesis and LC3B, BNIP3 and NIX for the mitophagy in IL-17RB overexpressing PSCs (IL-17RB OE) as well as the reverse effects due to the knockdown are not supported by the figures. Since the WB experiments are done in triplicate a quantification of the bands among all the experiments, showing standard deviation and statistical significance must be provided to support the conclusion described in the manuscript especially for those protein for which the authors claimed an altered expression.

We have removed the statements on the expression of CAV-1 and CCL-5 and believe that this does not change the overall message of the paper. However, we see a difference in IL-6 expression and have supported this with a quantification of the Western blots.

We reduced Figure 2C to show only P-MFF, Drp1, and MFN2. We removed the ambiguous results. We reduced Figure 2E to show the blots for PRK8, Beclin1, and p62. We inserted the correct interpretation in the main text. We removed figure 2D to concentrate on the main conclusion of the manuscript.

The main problem with a T-test is that there is no way to check the underlying normal distribution with such a small amount of data. With 3 observations per group, the p-value of the Wilcoxon rank-sum test cannot become smaller than .1, so this test also makes no sense. 

We, therefore, plotted the data. We decided against boxplots because with only three points, the median will be the mean and the quantiles cannot be estimated accurately. We have presented the data in a strip chart in the attached file “Western blot quantification.pdf”. We have prepared these graphics for the reviewer and have not included them in the manuscript in order not to exceed its scope.

Moreover, no results were shown about the efficiency of overexpression or knockdown for IL-17RB. Please provide qPCR and/or western blot of IL-17RB expression.

We added the requested Western blot to figure 2A.

One of the strongest effects that the authors described is on mitochondrial fission and mitophagy. Please provide more evidence about Drp1 release from mitochondria and accumulation in the cytoplasm in the IL-17RB OE and KD by cell fractionation or immunofluorescence with all appropriate control markers.

Due to the short time we have to answer the reviewers' questions, we are not able to perform the requested experiments. We will have to do these in continuing work.

Also, in figure 3 there is an evident discrepancy between the vehicle only controls. Please explain why the empty vectors have such effect on OCR too as was previously mentioned above for Fig2. This issue greatly impacts on the interpretation of the results. Please show the non-infected cells too, to establish if the empty vectors impact on the cellular functions discussed in this research and repeat all the relative experiments with a more suitable vector and a non-infected control.

Please see the answer above, which explains the effect of the empty vectors with the different backbones. Due to the short time for the revision, we are not able to generate new vectors.

Multiple unrelated PDAC cell lines were used for the presented assays in Figure 4 and 5, in particular the Hexokinase/STAT3 Western blot is shown in MiaPaca2 extracts rather than in the cell lines used for sphere formation (L3.6pl and HPAF-II) or conditioned media experiments. Different cell lines imply different phenotypes. Please provide a coherent set of experiments in terms of used cell lines and explain why the authors specifically used MIA PaCa2 cells for the in vivo experiments.

In a previous study, MiaPaCa-2 cells were co-injected with PSCs (Nature. 2016 Aug 25;536(7617):479-83.). MiaPaCa-2 formed structured pancreatic cancer tissue. We used MiaPaCa-2 cells because they have been used similarly in the literature and because they model pancreatic cancer in vivo better than other cell lines.

We used L3.6pl and HPAF-II in the sphere formation assay because we could not obtain spheres when employing MiaPaCa-2 in standard medium (DMEM-f12). The sphere formation assay yielded only cell clusters with MiaPaCa-2 cells.

In figure 4a, as for figure 1A, there is no evidence about the relative composition of xenografts: an immuno-histological assessment of PDAC and PSC would better clarify this issue.

We added the immunohistological assessment of PDAC and PSC to (the new) figure 4

The presented WB in Figure 4C is of very poor quality, and does not reflect the stated observation on phosphorylation/expression changes. The authors should provide more clear data to justify their claims. Also, in figure legends: there is no biological replicate nor any SD to be shown in this piece of data, please correct.

We show a quantification of the western blot in the attached power point file “Western blot quantification.pdf”.

Growth rates in Fig4E seem to be definitely affected by the Empty vector (VEC KD) alone when compared to control cells. Please justify this difference or repeat the experiments with more suitable vector, as previously noted in other figures.

Please see the above explanation of the differences between different backbones. Due to the short time for the revision, we are not able to generate new vectors. Moreover, we believe that other vectors would behave similarly.

A simple widefield acquisition of the co-staining shown in Figure 6A is not sufficient to confirm the localization of IL-17RB in mitochondria. A high z-stack resolution by confocal microscopy is required for an accurate subcellular localization.

We are currently unable to perform Z-stack immunofluorescence. Therefore, in the manuscript, we only present the possibility of co-expression. Follow-up experiments are planned that will shed more light on the role of intracellular receptor location and cytokine signaling.

Figure 6B:  these western blots are of very poor quality and misleading. In the left panel, beta-tubulin and Tom20 signals do not exclude each other in the different lanes showing a very poor enrichment for the RLT-PSC. Moreover, in the right panel is not clear which cell line has been analyzed and, again, why the IL-17RB level between the VEC controls are so different between them and compared to the left panel.

Even though in the left panel the signals of beta-tubulin and Tom20 are not mutually exclusive, mitochondrial enrichment of IL-17RB can be seen. This is all that our manuscript claims. We believe that mitochondria express the IL-17B receptor. This can be seen even better in L3.6pl tumor cells, whose IL-17RB expression cannot be explained by contamination of cellular components. The right panel shows the transfected PSCs. We have changed the text and caption to highlight this more clearly. You can see again the effect of the empty knockdown vector. We explain this with the different backbones.  Nevertheless, the effect of IL-17RB knockdown and overexpression can be seen in the distribution of the IL-17RB receptor on the mitochondria. To exclude contamination by cellular components, more elaborate experiments are necessary, which are currently being carried out. These still need some time, so they cannot be used for this manuscript. However, we think that the statements made here are sufficiently supported by the experiments.

MINOR POINTS

The introduction about pancreatic cancer is poorly written. It is very generic with some concepts not well explained: e.g. pancreatic cancer is not the most malignant tumor today but it sets to become the second most fatal cancer by 2030 (Latenstein, A.E.J., et al., Nationwide trends in incidence, treatment and survival of pancreatic ductal adenocarcinoma. Eur J Cancer, 2020. 125: p. 83-93). Please adjust and expand this part of the introduction.

We revised the beginning of the introduction according to the reviewer’s suggestions.

Please provide reference for “fatty acid synthesis and mitochondrial fission maintain pluripotency”.

The reference is EMBO J. 2017 May 15; 36(10): 1330-1347, but due to a request from another reviewer, we have revised and deleted this part.

Please indicate clearly in the figure 2F the alterations described in the results.

We have added arrows to indicate the results described in Fig. 2F.

Please display all barplots with the individual data points as shown in figure 6D.

As far as possible, we have converted the barplots to dotplots.

Numbering of the figure panels should be coherent with the order cited in the main text.

We revised the text and adjusted the ordering of the cited figure panels.

Please explain or speculate why you see an effect on IL6 or proliferation on OE sample only.

We removed this figure to make the manuscript more consistent due to a request of a different reviewer.

Please check time of administration of respiratory inhibitors because they do not coincide with the changing in OCR/ECAR graphs (Fig3, Fig5B).

We adjusted the graphs with the right time of administration of respiratory inhibitors.

Figure 4F: please provide reference to the sentence “cxcr4 expression contributing to tumour growth invasion and metastasis”

We added the reference Oncol Lett. 2018 Feb; 15(2): 1771–1776. to the manuscript.

In the immunofluorescence images in Fig1, the results could be clearer if each staining is shown as separate channels in addition to the merge image (as in Fig6) and if a higher magnification insert is added to the panel as for the IHC data. In addition, Figures 1 and 6 should display a clear and consistent nuclear counterstaining that for example is missing in Fig1 and inconsistent in Fig6.

We did not perform nuclear counterstaining for Fig 1. and cannot add it now. Without nuclear counterstaining, splitting the individual channels does not add any value. Since the immunofluorescence staining does not present extra value, we removed it.

Figure 7A should show the IL-17B expression of all the stromal cell clusters identified in the single-cell RNAseq data. Please provide the information of stellate cells as well.

Figure 7A was made using a public database. The stromal cells were not subdivided into further populations in this database. The stellate cells were not considered separately.

Line 392: OSC instead of PSC

We corrected the error.

Reviewer 4 Report

In this manuscript, Li et al  has demonstrated a major conclusion that the pancreatic stellate cells (PSCs) stimulate tumor progression through IL-17RB signaling. The best data that supports their hypothesis is the MiaPaCa-2 xenografts and the formation of spheroids. However, the manuscript suffers a major drawback that most of the data doesn’t match their hypothesis and sometimes misinterpreted in the results section. Due to this the manuscript cannot be accepted at the current form. If the authors respond to the following comments, it might be considered for publication.

Major Comments

  1. The western blots in figure 2 do not match with the description in results section. Overall the western blot data is not matching with the conclusions made by the authors. For example, the authors have mentioned that IL-17RB overexpression in PSCs decreases CCL-5, IL-6 and CAV-1 (Line 281-283). But, there is no prominent change based on western blotting. Again, Fig, 2C, 2E were not interpreted appropriately in the results section would be inclined not to show these data, or selectively show the blots that are convincing. Because, removing these data doesn’t change the overall conclusion of the manuscript.

  1. The authors have used MT assay in Fig. 2H and Fig.4E as an end point for cell proliferation. But, MTT data would not always correlate with the proliferation rate. This experiment is more suitable for viability assay. The authors should show the rate of proliferation wither by counting the cells ver the period of time or other suitable assays. But, MTT is the suitable approach.

  1. Again in Fig. 4C, the western blots do not completely match with the description in results. I would recommend to either remove the blots or change the explanation in results.

  1. In figure 5, the authors have explained that the oxygen consumption rate (OCR) of PANC-1 cells increased following the addition of culture media (CM) from IL-17BR OE PSCs. However, its not clear why the CM from Vehicle KD increases OCR while the CM from Vehicle OE des not show any difference as compared to the naïve cells (Fig. 5A). Shouldn’t be both the empty vectors yield the same result ? Moreover, the authors have mentioned the increase of OCR by vehicle KD in the results section. What does that actually mean?

  1. In figure 5B, the column graph for compensatory glycolysis from PANC-1 control vs PANC-1+CM VEC OE looks exactly same as the PANC-1 control vs PANC-1+CM+VEC KD. Please make sure that these graphs are not duplicated.

Minor Comments:

In figures 2 and 4, the order of individual panels does not exactly match the order they were cited in the text. It needs to be re-arranged. Also, fig.7G is cited way before the other panels in figure 7.

Overall the manuscript supports the conclusion that IL-17BR expression promotes tumor growth. The authors need to just show a cell proliferation data instead of MTT assay, which would suffice. On the metabolic pathways and mitochondrial stuff, most of the data do not match with their interpretation. Those data need to removed or the explained appropriately.

Author Response

Reviewer 4

Major Comments

  1. The western blots in figure 2 do not match with the description in results section. Overall the western blot data is not matching with the conclusions made by the authors. For example, the authors have mentioned that IL-17RB overexpression in PSCs decreases CCL-5, IL-6 and CAV-1 (Line 281-283). But, there is no prominent change based on western blotting.

Again, Fig, 2C, 2E were not interpreted appropriately in the results section would be inclined not to show these data, or selectively show the blots that are convincing. Because, removing these data doesn’t change the overall conclusion of the manuscript.

We took the reviewer's advice and removed the expression of CAV-1 and CCL-5, which is irrelevant for the conclusion of the manuscript. However, for IL-6, we see the relevance for the message of the paper. In our opinion, there is a difference in IL-6 expression.

We substantiated this with the quantification of the Western blots (see attached file “Western blot quantification.pdf”. We have prepared these graphics for the reviewer and have not included them in the manuscript in order not to exceed its scope).

Together with the ELISA analysis (Fig. 2G), we see a robust result.

We follow the suggestions of the reviewer to show only the relevant blots. We reduced Figure 2C to show only P-MFF, Drp1, and MFN2 and adjusted the figure legend (lines 333-334). Here, we see less P-MFF expression and more DRP1 expression comparing IL-17RB OE PSCs with VEC OE PSCs.

We substantiate this with quantification of the Western blots, and did not remove these results (lines 304-310). We thought about how to present the quantifications. The main problem with a T-test is that there is no way to check the underlying normal distribution with such a small amount of data. With 3 observations per group, the p-value of the Wilcoxon rank-sum test cannot become smaller than .1, so this test also makes no sense.

We, therefore, plotted the data. We decided against boxplots because with only three points, the median will be the mean and the quantiles cannot be estimated accurately. We have presented the data in a strip chart in the attached power point file “Western blot quantification.pptx”. We have prepared these graphics for the reviewer and have not included them in the manuscript in order not to exceed its scope.

We reduced Figure 2E to show the relevant blots and inserted the correct interpretation in the main text (lines 301-304). We removed figure 2D to concentrate on the main conclusion of the manuscript.

  1. The authors have used MT assay in Fig. 2H and Fig.4E as an end point for cell proliferation. But, MTT data would not always correlate with the proliferation rate. This experiment is more suitable for viability assay. The authors should show the rate of proliferation wither by counting the cells ver the period of time or other suitable assays. But, MTT is the suitable approach.

We used a dedicated MTT assay to evaluate cell proliferation. The cell culture starts with cell confluency of 20-30% and ends with cell confluency of 80-90%. This ensures that cell proliferation and not cell viability are analyzed.

  1. Again in Fig. 4C, the western blots do not completely match with the description in results. I would recommend to either remove the blots or change the explanation in results.

The reviewer is correct that the STAT3 Western blot in Figure 4C does not match the description in the results. We have removed this part from the results in line 379 as suggested by the reviewer. Correspondingly, we have adjusted the figure legend in lines 415-418.

  1. In figure 5, the authors have explained that the oxygen consumption rate (OCR) of PANC-1 cells increased following the addition of culture media (CM) from IL-17BR OE PSCs. However, its not clear why the CM from Vehicle KD increases OCR while the CM from Vehicle OE des not show any difference as compared to the naïve cells (Fig. 5A). Shouldn’t be both the empty vectors yield the same result ? Moreover, the authors have mentioned the increase of OCR by vehicle KD in the results section. What does that actually mean?

Backbone plasmids have a fundamental impact on cell gene expression (Cold Spring Harb Protoc. 2020 Aug 3;2020(8):095513.; Genes (Basel). 2020 Mar 10;11(3):291.). Different backbones induced different tag proteins, changing the downstream protein production. In this study, different backbones were used for the overexpression (OE) and knockdown (KD) of IL-17RB for technical reasons. Thus, we explain the changes in protein expression, especially of the empty knockdown vector (VEC KD) that the reviewer correctly observed. In the experiments, only the VEC OE group can be compared with the OE group and the VEC KD group with the KD group. Comparison of the VEC OE and VEC KD groups is not reasonable because of the different backbones. We have clarified this in the Materials and Methods section and have gone through the results section again. We have improved the parts that do not take this into account.

  1. In figure 5B, the column graph for compensatory glycolysis from PANC-1 control vs PANC-1+CM VEC OE looks exactly same as the PANC-1 control vs PANC-1+CM+VEC KD. Please make sure that these graphs are not duplicated.

We checked the graphs and made sure that they are no duplicates.   

Minor Comments:

In figures 2 and 4, the order of individual panels does not exactly match the order they were cited in the text. It needs to be re-arranged. Also, fig.7G is cited way before the other panels in figure 7.

We revised the text and adjusted the ordering of the cited figure panels.

Round 2

Reviewer 1 Report

In the revised manuscript entitled: IL-17B/RB Activation in Pancreatic Stellate Cells Promotes Pancreatic Cancer Metabolism and Growth, the authors picked up many suggestions from the review. I acknowledge that this version is substantially improved. Especially ambiguities raised by short and inadequately described experimental setups are now understandable and conclusive.

Unfortunately, the discussion represents the weakest chapter. Considerable parts of the discussion constitute of numerations of single sentences summarizing individual publications without conclusion or rounded summary.

The discussion should be carefully revised by a senior scientist.

Minor points:

Are the immunofluorescence pictures produced by confocal microscopy (as suggested in the acknowledgements) or by standard microscopy as suggested in Material and Methods?

The in-text citation of literature is not consistent. Numbered in most of the text but in the new parts numbered or cited by journal name and volume.

There are some spelling errors left.

Author Response

Major points:

Unfortunately, the discussion represents the weakest chapter. Considerable parts of the discussion constitute of numerations of single sentences summarizing individual publications without conclusion or rounded summary.

The discussion should be carefully revised by a senior scientist.

We rewrote the discussion to focus on two major points. First, we discuss the effects of stroma-tumor interaction on tumor progression, and second, the influence of the stroma on oxidative phosphorylation and glycolysis.

Minor points:

Are the immunofluorescence pictures produced by confocal microscopy (as suggested in the acknowledgements) or by standard microscopy as suggested in Material and Methods?

Figure 1C was made with standard microscopy, and figure 6A was made using confocal microscopy. We added to the main text, the materials and methods section, and the legend of figure 6A to make this clearer.

The in-text citation of literature is not consistent. Numbered in most of the text but in the new parts numbered or cited by journal name and volume.

It was technically impossible to insert the references as numbers in brackets in the manuscript. We have therefore included the references as plain text until we fix this problem. We are in touch with the editors to do so.

There are some spelling errors left.

We checked the spelling throughout the document and corrected the errors we found.

Reviewer 3 Report

Although I appreciate the efforts the authors made to address the reviewer’s comments, most of the critical experiments needed to sustain the author’s claims were not performed.

The explanation provided for Figure 1A are really generic. The Immunofluorescence on tissue has just been removed and the IHC does not clearly show the coexpression of aSMA with IL17RB in the stromal compartment. Moreover, the staining does not match the single cell data, now reported in the fig1A, where many cell types do not express IL17RB while the IHC shows expression in almost all cells depicted. The conclusions presented in Fig1A, at least for the stromal compartment, are not supported by the data and more controls should be performed to use the IHC experiments as proof.

In addition, there is not any details on the results presented in the fig1D (what do the dots represent? Why and how the expression level was scaled between 0 and 1?). The data refer to Dominguez et al paper, but in that study is stated that the human data were reanalyzed form Peng et al. In the present manuscript the authors refer to PRJCA001063 that is the Peng paper. Those data are publicly available and they do contain the stellate cells, so it is not clear the author’s response. Did the authors use the reanalyzed data or the original data? Because it seems that the results presented are from mouse samples from Dominguez manuscripts and not from human PDAC as stated in the methods and figure legend.

The materials and methods section, as specified by the guidelines of the journal, should provide enough details to allow other researchers to reproduce the results.

The references mentioned for the effects of different viral backbones are just principles of genetic engineering and do not provide any justifications to the discrepancies observed. The reviews described how the different vector backbones impact on the expression of the cloned gene/sequence of interest; they do not show or justify that different empty lentiviral backbones produce not specific effects on genes that are the targets of the study.

I understand the technical issues, but these discrepancies hamper most of the conclusions and interpretation of the data.

In addition, even the non-transduced controls show very different IL17RB levels among the WBs shown (eg. Fig 1G and 1H) revealing inconsistency among the experiments and many images are excessively manipulated/exposed to increase the contrast (Fig1G, Fig.2A, 2G).

The experiments with the recombinant IL-17B are unclear. Do the authors suggest that IL17B/RB axis function only if the IL17B is administered through exosome?

The results described in Fig2 are inconsistent and conclusions made are not supported by the data. The upregulation or downregulation of alpha SMA is inconsistent as shown by the supplementary materials. In Fig 1G the alpha SMA upregulation seemed more convincing and the authors claimed no differences. Authors claimed IL6 upregulation of IL-6 protein level, but no effect was shown on secretion. Authors claimed other differences or no change in protein levels but quantification shown in supplementary materials are highly variable and no clear conclusions can be made from those experiments (I suppose that in the supplementary materials there are some mistakes in the labels since some graphs do not match the WB images).

The explanation regarding the different cell lines used are not convincing and some assumptions are simply not true. MiaPaca2 does not better recapitulate the human PDAC; as it is largely showed in many publications MiaPaca2 represent only grade 3 PDAC or QM-PDAC (Collison et al, Nature Med 2011). In addition, Fig4 does not support the claim that the co-injection of PSCs and MiaPaca2 cells resulted in tumor formation with stroma enveloping the tumor cells as it is also clearly showed in the referenced Sousa’s paper.

Author Response

The explanation provided for Figure 1A are really generic. The Immunofluorescence on tissue has just been removed and the IHC does not clearly show the coexpression of aSMA with IL17RB in the stromal compartment.

Moreover, the staining does not match the single cell data, now reported in the fig1A, where many cell types do not express IL17RB while the IHC shows expression in almost all cells depicted. The conclusions presented in Fig1A, at least for the stromal compartment, are not supported by the data and more controls should be performed to use the IHC experiments as proof.

We made additional IL-17RB / a-SMA double stainings and examined random image sections of the stroma. In the attached file: "immunohistochemistry.pdf" an image section is shown as an example (A). We used ImageJ to perform color deconvolution to separate the individual stainings (B and C). We layered them semitransparent (33%) on top of each other again (D). One can now clearly see the co-expression of IL-17RB and a-SMA of various stromal cells. As an example, we have marked one with a black arrow.

We think the reviewer means figure 1C when referring to the single-cell data. In the bottom row, human PSCs are shown, where after stimulation almost all cells are IL-17RB positive. The top row shows mouse data, which is not comparable to the human data. But there are IL-17RB positive murine PSCs. Consistent with the human single-cell data, almost all cells are IL-17RB or a-SMA positive in the activated stroma of pancreatic cancer. This is consistent with publicly available data from the Human Protein Atlas, which also shows almost complete positivity of pancreatic cancer stroma for IL-17RB and a-SMA in single stainings (see the second image in the file: "immunohistochemistry.pdf").

In addition, there is not any details on the results presented in the fig1D (what do the dots represent? Why and how the expression level was scaled between 0 and 1?). The data refer to Dominguez et al paper, but in that study is stated that the human data were reanalyzed form Peng et al. In the present manuscript the authors refer to PRJCA001063 that is the Peng paper. Those data are publicly available and they do contain the stellate cells, so it is not clear the author’s response. Did the authors use the reanalyzed data or the original data? Because it seems that the results presented are from mouse samples from Dominguez manuscripts and not from human PDAC as stated in the methods and figure legend.

Single-cell RNAseq data was used to determine IL-17B expression. Expression levels were normalized to the interval from 0 to 1. Each dot in Fig. 1D represents a single cell with IL-17B expression. PSCs are contained in the fibroblast compartment in healthy tissue and differentiate into CAFs in the tumor environment. In the first two columns, fibroblasts from healthy tissue were compared with CAFs. We reanalyzed the original human database PRJCA001063 according to the analysis method described by Dominguez et al. We have added the Material and Methods section and to the figure legend to make this clearer.

The materials and methods section, as specified by the guidelines of the journal, should provide enough details to allow other researchers to reproduce the results.

We revised the Materials and Methods sections to make the experiments more comprehensible.

The references mentioned for the effects of different viral backbones are just principles of genetic engineering and do not provide any justifications to the discrepancies observed. The reviews described how the different vector backbones impact on the expression of the cloned gene/sequence of interest; they do not show or justify that different empty lentiviral backbones produce not specific effects on genes that are the targets of the study.

I understand the technical issues, but these discrepancies hamper most of the conclusions and interpretation of the data.

We think that a backbone may have specific effects, e.g., in protein expression. We did not find a reference for this, but the phenomenon is discussed on the internet (https://www.researchgate.net/post/Why_does_shRNA_Empty_vector_show_reduced_RNA_and_protein_expression). We consider it is scientifically justified to compare only the VEC OE group with the OE group and the VEC KD group with the KD group. The statements of the paper rely mostly on the comparison of the VEC OE group with the OE group. No cells transfected with different backbones are compared. To definitively resolve discrepancies, RLT-PSCs would need to be transfected with new vectors and all experiments repeated.

These experiments cannot be performed within the time given for the completion of the manuscript. If the reviewer considers these experiments to be mandatory, she or he should consider rejecting the whole work.

In addition, even the non-transduced controls show very different IL17RB levels among the WBs shown (eg. Fig 1G and 1H) revealing inconsistency among the experiments and many images are excessively manipulated/exposed to increase the contrast (Fig1G, Fig.2A, 2G).

In Fig. 1G, the non-transduced controls were stimulated with rhIL-17B. We interpreted these experiments as showing no significant differences in IL-17RB expression between unstimulated cells and PSCs stimulated with 200 or 400 ng/ml rhIL-17B. In Fig. 1H, we stimulated PSCs with exosomes. Exosomes from L3.6pl and Panc-1 pancreatic cancer cells induced IL-17RB expression in PSCs. These experiments consistently showed at least a partial effect of Exosomes in regulating IL-17RB expression in PSCs.

The experiments with the recombinant IL-17B are unclear. Do the authors suggest that IL17B/RB axis function only if the IL17B is administered through exosome?

Recombinant human IL-17B does not significantly upregulate IL-17RB (Fig. 1G). In conjunction with the IL-17RB-inducing effect of tumor-derived IL-17B carrying exosomes (Fig. 1H), we believe that exosomes have at least a partial effect in regulating IL-17RB. Therefore, we removed most of the experiments with recombinant IL-17B from the previous version of the manuscript. These experiments played only a minor role in the main conclusions of the paper.

The results described in Fig2 are inconsistent and conclusions made are not supported by the data. The upregulation or downregulation of alpha SMA is inconsistent as shown by the supplementary materials. In Fig 1G the alpha SMA upregulation seemed more convincing and the authors claimed no differences.

We changed Fig. 1G to state that α-SMA did change upon stimulation with very high IL-17B concentrations (400 ng/ml) that are not found in vivo.

Authors claimed IL6 upregulation of IL-6 protein level, but no effect was shown on secretion.

Fig. 2E shows that IL-17RB overexpression significantly decreases IL-6 secretion, while IL-17RB knockdown induces no changes.

Authors claimed other differences or no change in protein levels but quantification shown in supplementary materials are highly variable and no clear conclusions can be made from those experiments (I suppose that in the supplementary materials there are some mistakes in the labels since some graphs do not match the WB images).

The reviewer is right about the variance of the quantifications. We reviewed the supplementary data again but could not find the quantifications that don't match the WB images. Though the tendency in each graph fits our claims, the statistics of the quantifications are not always significant. We have now explicitly stated in the result section that we reported trends when statistical significance was not reached.

The explanation regarding the different cell lines used are not convincing and some assumptions are simply not true. MiaPaca2 does not better recapitulate the human PDAC; as it is largely showed in many publications MiaPaca2 represent only grade 3 PDAC or QM-PDAC (Collison et al, Nature Med 2011).

The reviewer is right in stating that MiaPaCa-2 cells represent only grade 3 PDAC or QM-PDAC. We selected MiaPaCa-2 cells mainly because they have already been used with PSCs in the literature [Nature. 2016 Aug 25;536(7617):479-83.]. We aimed to minimize risks in the experimental design. Since co-injection of MiaPaCa-2 cells and PSCs is already described in the literature, we estimated the risk of the experiment to be low with this cell combination.

In addition, Fig4 does not support the claim that the co-injection of PSCs and MiaPaca2 cells resulted in tumor formation with stroma enveloping the tumor cells as it is also clearly showed in the referenced Sousa’s paper.

We removed the claim that the co-injection of PSCs and MiaPaca2 cells resulted in tumor formation with stroma enveloping the tumor cells.

Reviewer 4 Report

The authors have responded to all the comments. They have revised the manuscript appropriately. 

Author Response

Nothing to be done here

Round 3

Reviewer 3 Report

I really appreciate all the efforts the authors made to answer the reviewer’s criticisms but without performing the critical experiments the conclusions are not supported by the data. The experiments showed only trends with many technical issues and variability so the observations seem just circumstantial.

For example I do not understand how the authors can claim that “there was no change in Beclin1-expression indicating uncompromised macro authophagyt” and “we observed decreased PRK8 expression” when the WB pictures and the quantification data, that the authors claimed is properly labelled, showed a very similar “trend” where the changes in Beclin expression seems more convincing than PRK8.

Also, the answer for the criticism regarding the single cells data seems inaccurate: “Each dot in Fig. 1D represents a single cell with IL-17B expression”. In the graph in figure 1D only 14 dots are shown. If the data is scaled between 0 and 1 and the dots represents only those single cells among the 5769 CAFs reported that have a value higher than 0.2, it means that most of the population of CAF displayed no changes in IL-17B expression.

I think that more convincing experiments need to be performed for the claims described in the manuscript.